

# Impact of a nitrogen emission control area (NECA) on the future air quality and nitrogen deposition to seawater in the Baltic Sea region

Matthias Karl[1], Johannes Bieser[1], Beate Geyer[1], Volker Matthias[1], Jukka-Pekka Jalkanen[2], Lasse Johansson[2], and Erik Fridell[3]

[1]Institute of Coastal Research, Helmholtz-Zentrum Geesthacht, 21502 Geesthacht, Germany.
[2]Atmospheric Composition Research, Finnish Meteorological Institute, P.O. Box 503 FI-00101 Helsinki, Finland.
[3]IVL Swedish Environmental Research Institute, P.O. Box 53021, SE 40014, Gothenburg, Sweden.

**Correspondence:** M. Karl (matthias.karl@hzg.de)

**Abstract.** Air pollution due to shipping is a serious concern for coastal regions in Europe. Shipping emissions of nitrogen oxides ($NO_X$) to air on the Baltic Sea are of similar magnitude ($330\,\text{kt}\,\text{y}^{-1}$) as the combined land-based NOX emissions from Finland and Sweden in all emission sectors. Deposition of nitrogen compounds originating from shipping activities contribute to eutrophication of the Baltic Sea and coastal areas in the Baltic Sea region. For the North Sea and the Baltic Sea a nitrogen

emission control area (NECA) will become effective in 2021; in accordance with the International Maritime Organization (IMO) target of reducing $NO_X$ emissions from ships. Future scenarios for 2040 were designed to study the effect of enforced and planned regulation of ship emissions and the fuel efficiency development on air quality and nitrogen deposition. The Community Multiscale Air Quality (CMAQ) model was used to simulate the current and future air quality situation. The meteorological fields, the emissions from ship traffic and the emissions from land-based sources were considered at a grid

resolution of $4 \times 4\,\text{km}^2$ for the Baltic Sea region in nested CMAQ simulations. Model simulations for the present-day (2012) air quality show that shipping emissions are the major contributor to atmospheric nitrogen dioxide ($NO_2$) concentrations over the Baltic Sea. In the business as usual (BAU) scenario, with the introduction of the NECA, $NO_X$ emissions from ship traffic in the Baltic Sea are reduced by about 80 % in 2040. An approximate linear relationship was found between ship emissions of $NO_X$ and the simulated levels of annual average $NO_2$ over the Baltic Sea in year 2040, when following different future

shipping scenarios. The burden of fine particulate matter ($PM_{2.5}$) over the Baltic Sea region is predicted to decrease by 35–37 % between 2012 and 2040. The reduction of $PM_{2.5}$ is larger over sea, where it drops by 50–60 % along the main shipping routes, and smaller over the coastal areas. The introduction of NECA is critical for reducing ship emissions of $NO_X$ to levels that are low enough to sustainably dampen ozone ($O_3$) production in the Baltic Sea region. A second important effect of the NECA over the Baltic Sea region is the reduction of secondary formation of particulate nitrate. This lowers the ship-related

$PM_{2.5}$ by 72 % in 2040 compared to present-day, while it is reduced by only 48 % without implementation of the NECA. The effect of a lower fuel efficiency development on the absolute ship contribution of air pollutants is limited. Still, the annual mean ship contributions in 2040 to $NO_2$, sulphur dioxide and $PM_{2.5}$ and daily maximum $O_3$ is significantly higher if a slower fuel efficiency development is assumed. Nitrogen deposition to the seawater of the Baltic Sea decreases on average by 40–44 % between 2012 and 2040 in the simulations. The effect of the NECA on nitrogen deposition is most significant in the western

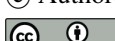



part of the Baltic Sea. It will be important to closely monitor compliance of individual ships with the enforced and planned emission regulations.

# 1 Introduction

Air pollution due to shipping is a serious concern for coastal regions in Europe (Viana et al., 2014; Matthias et al., 2010). Globally, nearly 70 % of the exhaust emitted from ship traffic occurs within a corridor of 400 km along the coastline (Endresen et al., 2003). Since emissions from ships can be transported in the atmosphere over several hundreds of kilometres, they have the potential to diminish the air quality in coastal areas. In addition to the primary emitted particles in the ship exhaust, secondary particles are formed in the atmosphere by oxidation of emitted gaseous precursors - nitrogen oxides ($NO_X$) and sulphur dioxide ($SO_2$) - during the dispersion of the ship exhaust. Mainly by contributing to the ambient levels of fine particulate matter, $PM_{2.5}$ (particles with diameter less than 2.5 μm), emission from ship traffic are responsible for a large number of premature deaths globally (Corbett et al., 2007). According to Sofiev et al. (2018b) the worldwide use of cleaner marine fuels with a lower content of sulphur will strongly reduce the ship-related premature mortality and morbidity, by 34 % and 54 % respectively. In northern Europe, the health-related external costs from international shipping in the Baltic Sea and North Sea are expected to decrease by 36 % between 2000 and 2020 (Brandt et al., 2013). This reduction is mainly a consequence of the introduction of the sulphur emission control area (SECA) for Baltic Sea (enforced 2005) and North Sea (enforced 2006), which step-wise reduced the sulphur content in ship fuels.

However, air emissions of $NO_X$ from ship traffic remained almost constant throughout the last decade, and the impact of $NO_X$ will remain a concern for health. Shipping emissions of $NO_X$ on the Baltic Sea are of similar magnitude as the combined land-based $NO_X$ emissions from Finland and Sweden in all emission sectors (Jalkanen and Stipa, 2009). While EU air quality legislation will lead to a decline of land-based emissions of $NO_X$ in the future, ship emissions - without more stringent emission control measures on $NO_X$ - will rise with the projected annual growth of maritime traffic in the Baltic Sea of about 5 % (Stipa et al., 2007). As a consequence the relative importance of shipping emissions compared to land-based emission sources of $NO_X$ is expected to increase. A review of model studies on ship emissions showed that $NO_X$ emissions from international shipping on European seas could be equal to land-based emission sources in Europe (EU-27) from 2020 onwards and confirmed that the contribution of the shipping sector to future air pollution in Europe will increase (EEA, 2013).

Through atmospheric oxidation exhaust emissions of $NO_X$ are converted to gaseous nitrous acid ($HNO_3$). This conversion of nitrogen dioxide ($NO_2$) to $HNO_3$ takes place at a rate of approximately 5 % per hour, causing an atmospheric lifetime of $NO_X$ of about 24 hours (Geels et al., 2012). $HNO_3$ is a sticky compound, which is, in the presence of ammonia ($NH_3$), converted by gas phase/particle partitioning to particulate nitrate ($NO_3^-$). Nitrate is removed from the atmosphere via dry and wet scavenging, contributing to deposition of oxidized nitrogen to the sea. Atmospheric deposition of nitrogen (N)-containing compounds play a role in the eutrophication of the coastal marine environment (e.g., Paerl, 1995). Eutrophication of the sea is caused by high inputs of nutrients (nitrogen and phosphorus) resulting in the production of algal blooms, followed by the accumulation of organic material which after sedimentation results in the depletion of oxygen in the bottom water of stratified



areas of the sea. Atmospheric deposition of nitrogen accounts for approximately one third of the total nitrogen input to the Baltic Sea (HELCOM, 2011).

Several studies have used atmospheric chemistry-transport models (CTM) to investigate the composition and fluxes of atmospheric nitrogen to the Baltic Sea basin (Hertel et al., 2003; Hongisto and Joffre, 2005; Langner et al., 2009; Hongisto, 2011; Bartnicki et al., 2011; Geels et al., 2012) mainly focusing on the influence of meteorological and climatological factors and the inter-annual variability of meteorological conditions. Annual atmospheric deposition of total nitrogen to the Baltic Sea basin computed with the CTM model EMEP/MSC-W (Simpson et al., 2012) declined between 1995 ($305\,\mathrm{kt\,y^{-1}}$) and 2015 ($222\,\mathrm{kt\,y^{-1}}$) by 27 % (Bartnicki et al., 2017; data normalised to inter-annual changes of meteorological conditions). While the deposition of oxidised nitrogen decreased by 35 % during this period, reduced nitrogen, i.e. mainly $NH_3$ and particulate ammonium ($NH_4^+$), decreased by only 12 % (Bartnicki et al., 2017). Based on atmospheric CTM calculations, it has been estimated that the atmospheric deposition of N-containing compounds originating from ship exhaust, depending on the season, can contribute to more than 50 % of the total atmospheric deposition of nitrogen in some areas of the Baltic Sea (Stipa et al., 2007).

Emissions from shipping are regulated globally by Annex VI "Regulations for the Prevention of Air Pollution from Ships" to the MARPOL Convention (IMO, 2008a). The $NO_X$ emission reduction scheme of IMO MARPOL Annex VI is based on the Tier standards as described in the $NO_X$ Technical Code (IMO, 2008b). Tier I, implemented in the year 2000; introduced emission standards for ships constructed between 1 January 2000 and 1 January 2011 up to 10 % stricter than those that applied for ships built before 2000. Tier II, implemented in 2011, enforced up to 15 % stricter standards than Tier I for ships constructed after 1 January 2011. Tier I and Tier II limits are worldwide and apply to all new marine diesel engines. The third regulation stage, Tier III, will only affect ships sailing inside the designated nitrogen emission control areas (NECA). A NECA for the Baltic Sea, North Sea and English Channel will become effective in 2021. In the following, we refer to the northern European NECA simply as "the NECA". From 1 January 2021 onwards, new built ships in the Greater North Sea and Baltic Sea have to comply with the stringent Tier III regulations for $NO_X$-emissions, approximately 75 % stricter than Tier II. To fulfil the requirements of Tier III, ship owners have to use abatement methods such as exhaust control technologies (catalyst converters, etc.) or use liquefied natural gas as fuel for new ships.

For the North Sea, Matthias et al. (2016) using a regional atmospheric CTM system and detailed shipping emission inventories for the present-day and future situations, estimated that upon introduction of the NECA in 2016, levels of $NO_2$, particulate nitrate and ozone ($O_3$) in 2030 would not change compared to the year 2011, because the growth in ship traffic compensates potential emission reductions. A delayed introduction of the NECA by 5 years (in 2021), would cause concentration increases of these pollutants by 10–15 % compared to today (Matthias et al., 2016). The study by Matthias et al. (2016) assumes an increase in ship number by 1 % p.a., an increase of transported cargo of 2.5 % p.a. and a ship renewal rate of 2.5 % p.a. independent of ship size. The study considered no gains in fuel efficiency of new built ships. Clearly, predicted consequences of the Tier III $NO_X$ emission regulation on future shipping emissions depend critically on the projected growth of transported volume, the increase in ship number and the share of new ships in the future fleet. In a similar study, Jonson et al. (2015) investigated the effect of the NECA introduced in 2016 on the air quality in 2030, assuming a moderate increase in ship activity. According to



their future scenario, total NO$_X$ emissions in the Baltic Sea and the North Sea will almost be unchanged in 2030 compared to 2010, if the NECA is not implemented. However, implementation of the NECA in 2016 will lead to significantly lower NO$_X$ emissions from ships in 2030, resulting in slight reductions in the burden on health due to shipping (Jonson et al., 2015). The emission study by Kalli et al. (2013), which calculates the emissions separately for every ship taking into account expected traffic growth and fleet renewal, corroborates the strong decrease of NO$_X$ shipping emissions (by 11 % in 2020 and by 79 % in 2040) when the NECA is established in 2016.

The goal of the present study is to investigate the effect of the implementation of the NECA in 2021 on the air quality in the Baltic Sea region and on the total deposition of nitrogen to the Baltic Sea in 2040. In addition to the effect of the NECA regulation, we also look into possible future developments which might diminish the beneficial effect of the NECA, such as failing to achieve increased fuel efficiency of ships.

Several future shipping emission scenarios for the year 2040 were designed. These scenarios were based on the projected development of the economic growth and ship traffic volume in accordance with the study by Kalli et al. (2013). Land-based emission sources are assumed to follow the emission reduction due to current EU legislation. Three cases with respect to future air quality were considered: (1) implementation of the NECA in 2021; (2) no implementation of the NECA; and (3) alternative assumptions for the fuel efficiency of the ship fleet in combination with NECA.

A regional atmospheric CTM system using the Community Multiscale Air Quality (CMAQ) model (Byun and Schere, 2006; Appel et al., 2013), similar to that used in the study by Matthias et al. (2016), was used to simulate the present-day and future air quality conditions in the Baltic Sea region. The advantage of the applied CTM system for the Baltic Sea compared to previous studies in the same region (Matthias et al., 2016; Jonson et al., 2015; Hongisto, 2014) is the higher spatial and temporal resolution of all components driving the chemistry-transport calculations. The meteorological fields, the emissions from ship traffic and the emissions from land-based sources were considered at a grid resolution of $4 \times 4 \, km^2$ for the inner-most model domain in the nested CMAQ runs. Higher resolution of shipping emissions, which are obtained based on ship positions acquired from 4-minute AIS (Automatic Identification System) records and detailed ship characteristics using the Ship Traffic Emission Assessment Model (STEAM; Jalkanen et al., 2009; 2012; Johansson et al., 2013; 2017) in combination with the higher resolution of the chemistry-transport computation allow for a better resolution of the individual ship's plumes. Moreover, the high resolution meteorology (0.025° grid) resolves convective precipitation, which is expected to improve the timing and amount of predicted rainfall, crucial for the determination of the nitrogen inputs to the Baltic Sea.

The focus of the present study will be on the computational model results for summer, defined as the average of the period June-August (JJA), when assessing the changes of air quality and deposition between the future scenarios and the present-day situation. In summer, emissions from shipping are highest and the photochemical conversion of the ship exhaust constituents into compounds that are readily scavenged by precipitation is faster than in other seasons. Therefore, ship-originated oxidised nitrogen deposition to the sea is highest during the summer (Hongisto, 2014). In addition, the seasonal variation of air quality indicators and of the accumulated nitrogen deposition to seawater is presented.

A first set of model runs was performed for the situation in year 2012. The present day model results on nitrogen deposition and the air quality situation is analysed. Modelled deposition of nitrogen was evaluated in two steps, first the predicted rainfall



amount and frequency is compared to daily precipitation measurements from rain gauge stations in Sweden, and second the wet deposition of oxidised and reduced nitrogen is compared against measurements of the "Cooperative Programme for Monitoring and Evaluation of the Long-range Transmission of Air Pollutants in Europe" (EMEP) programme. Present-day model results on air quality are evaluated with measurements from the regional background stations of the EMEP monitoring network in

the Baltic Sea region. A companion paper by Karl et al. (2018) presents a more detailed comparison of the model results for the current air quality situation with land-based observations of air pollutant concentrations in the Baltic Sea region. The contribution of shipping emissions to the modelled concentration of air pollutants was determined from the difference between a reference run that included all emissions and a "Noship" run that excluded emissions from ship traffic (zero-out method).

A second set of model runs was performed to assess the effect of projected emissions from shipping for the year 2040.

Future air quality and nitrogen deposition is analysed, in order to investigate: (1) the effect of establishing the NECA in 2021 compared to a future situation without NECA; and (2) the effect of a lower fuel efficiency increase than expected based on continuation of the current trend. Changes of the ship contribution to regulated air pollutants and to nitrogen deposition over seawater between the present-day simulation and the future scenario simulations are presented. Finally, recommendations with respect to the future regulations and their possible impacts and side-effects are given.

## 2   Chemistry-transport modelling

### 2.1   CMAQ model description

Regional chemistry/transport model simulations with the Community Multiscale Air Quality (CMAQ) model v5.0.1 (Byun and Ching, 1999; Byun and Schere, 2006; Appel et al., 2013; Appel et al., 2017) were performed to assess the effect of emissions from ship traffic on the present-day and future air quality of the Baltic Sea region. The CMAQ model computes the air

concentration and deposition fluxes of atmospheric gases and aerosols as a consequence of emission, transport and chemical transformation. The atmospheric chemistry of reactive species is treated by the Carbon Bond V mechanism (Yarwood et al., 2005), with updated toluene chemistry (Whitten et al., 2010) and chlorine radical chemistry (mechanism cb05tucl; Sarwar et al., 2012).

The aerosol scheme AERO5 is used for the formation of secondary inorganic aerosol (SIA). Aerosol growth and nucleation

is simulated by three lognormal distributed modes, each represented by three moments (Binkowski and Roselle, 2003). The Aitken and accumulation modes represent $PM_{2.5}$ and the coarse mode represents particulate matter with diameter >2.5 µm ($PM_{coarse}$). The instantaneous gas phase/aerosol equilibrium partitioning of sulphuric acid ($H_2SO_4$), $HNO_3$, hydrochloric acid (HCl) and $NH_3$ on the fine particle modes is solved with the ISORROPIA v1.7 mechanism (Nenes et al., 1999). Dynamic mass transfer is simulated for the coarse particle mode because large particles often do not reach equilibrium with the gas phase for

typical atmospheric time scales (Meng and Seinfeld, 1996). For the coarse mode, semi-volatile inorganic species are allowed to condense and evaporate, while $H_2SO_4$ does not evaporate again from the coarse mode. Because of the dynamic mass transfer to coarse particles it is possible to use CMAQ for the simulation of chloride ($Cl^-$) replacement by $NO_3^-$ in mixed marine/urban air masses (Foley et al., 2010) which could be an important aerosol process in the Baltic Sea region.



Sea salt emissions were calculated inline by the parameterization of Gong (2003), as described in Kelly et al. (2010). Sea salt surf zone emissions were deactivated because of considerable overestimations in some coastal regions (Neumann et al., 2016b). The formation of secondary organic aerosol (SOA) from isoprene, monoterpenes, sesquiterpenes, benzene, toluene, xylene, and alkanes (Carlton et al., 2010; Pye and Pouliot, 2012) is included. SOA formation pathways include the traditional two product representation, reaction of volatile organic compounds (VOC) to give non-volatile products, oxidative ageing of primary organic aerosol, acid-catalysed enhancement of SOA mass, oligomerization reactions and in-cloud aqueous-phase oxidation.

Three types of clouds are modelled in CMAQ: sub-grid convective precipitation clouds, sub-grid non-precipitating clouds and grid-resolved clouds. The meteorological model provides information about the grid-resolved clouds and CMAQ subsequently does not apply further cloud dynamics for this cloud type. For the two types of sub grid clouds, the cloud module in CMAQ vertically redistributes pollutants, calculates in-cloud and precipitation scavenging and performs aqueous phase chemistry calculations. Sub-grid clouds are only simulated in CMAQ when the meteorological driver uses a convective cloud parameterization. Hence no sub-grid clouds are treated for the $4 \times 4 \, km^2$ model domain because the convective clouds are resolved by the meteorological model.

Wet deposition of gases and particles is computed by the resolved cloud model of CMAQ which estimates how much certain vertical model layers contributed to the precipitation. The precipitation flux for each model layer is computed as a function of the non-convective precipitation rate, the sum of hydrometeors (rain, snow, and graupel) and the layer thickness (see Foley et al. (2010) for details).

Dry deposition is determined as the product of the atmospheric concentration and the deposition velocity. The dry deposition velocity is modelled in CMAQ using the resistance analogy, where resistances are defined along pathways from the atmosphere to the surface which act in parallel or in series. Details on the deposition pathways in CMAQ can be found in Pleim and Ran (2011). The deposition velocity for particles is calculated based on the aerosol size distribution, as well as meteorological and land-use information. For large particles, the dry deposition transfer is by turbulent air motion and by direct gravitational sedimentation. The dry deposition algorithm for particles includes an impaction term in the coarse mode and the accumulation mode.

In the resistance method it is assumed that the surface concentration of the chemical species is zero. However, $NH_3$ can be both emitted from and deposited to surfaces depending on its atmospheric concentration. This bi-directional nature of the air-surface exchange can modify the atmospheric transport and environmental impact of ammonia. Bi-directional fluxes of $NH_3$ over marine surfaces have been documented in a review by Hertel et al. (2006). In fact, inclusion of the bi directional air-water exchange in a CTM resulted in lower overall dry deposition of $NH_3$ to coastal waters (Sorensen et al., 2003). However, until now, the parameterization of the bi-directional flux has not been evaluated to a large extent for marine waters. Although the bi-directional flux of $NH_3$ is implemented in CMAQ v5.0.1, the option was not used in this study. Because we are mainly interested in the differences of total nitrogen deposition due to changes in emission alone, the outcome of this study will be less affected by the sensitivity of the modelled nitrogen deposition to bi-directional fluxes of ammonia.



## 2.2 Setup of the model

Nested simulations with CMAQ were performed on a horizontal resolution of $4 \times 4\,\mathrm{km}^2$ to simulate the current and future air quality situation for the entire Baltic Sea region. The model was set up on a $64 \times 64\,\mathrm{km}^2$ grid for entire Europe, subsequently on an intermediate nested $16 \times 16\,\mathrm{km}^2$ grid for Northern Europe, and finally on two nested $4 \times 4\,\mathrm{km}^2$ grids, one for the southern

Baltic Sea (Baltic major) and one for the northern Baltic Sea (including Bothnian Bay and Gulf of Finland). The nesting is visualized in Fig. 1a and the geographic details of the high resolution domain is shown in Fig. 1b. The vertical dimension of the model extends up to 100 hPa in a sigma hybrid pressure coordinate system with 30 layers. Twenty of these layers are below approximately 2 km; the lowest layer extends to ca. 36 m above ground. A spin-up period of one month (December 2011) was used for the initialization of the model runs, sufficiently long to prevent that initial conditions have an effect on the simulated

atmospheric concentrations of the investigated period (year 2012).

## 2.3 Meteorological fields

The meteorological fields that drive the CTM were simulated with the COSMO-CLM, version 5.0, for the year 2012 (Geyer, 2014) using the ERA Interim reanalysis and spectral nudging technique to force the model. COSMO itself is the operational weather forecast model applied and further developed by a consortium of national weather services whereas COSMO-CLM

stands for the climate mode used and developed by the limited area modelling community (clm-community; Rockel et al., 2008).

   The meteorological runs were performed first on a $0.11 \times 0.11$ degrees rotated lat-lon grid using 40 vertical layers up to 22 km for entire Europe. The output was used as forcing of a high-resolution nested meteorology run on a $0.025 \times 0.025$ degrees grid; 50 vertical levels were used for this simulation for the Baltic Sea region. The convection permitting configuration is used on

the high-resolution grid, e.g. only shallow convection is based on Tiedtke scheme, resolving convective precipitation clouds. The meteorological fields were processed afterwards using a modified version of CMAQ's Meteorology Chemistry Interface Processor (MCIP; Otte and Pleim, 2010) to match the extension, resolution and projection of the CMAQ nested grids.

   Based on the temperature anomalies and precipitation anomalies for the decade 2004–2014 for Baltic Proper, the year 2012 was chosen as meteorological reference year for the CTM simulations. Year 2012 anomalies for 2 m temperature ($\pm 2\,^\circ\mathrm{C}$)

and total precipitation ($\pm 25\,\mathrm{mm}$) were closely aligned to the decadal average of the 2004–2014 period. The meteorological year 2012 was also used in CTM calculations of the future air quality situation to avoid complication of the interpretation of changes between present-day and the future. Hence, future changes of the air quality are solely due to changed land-based and shipping emissions.

## 2.4 Boundary conditions

The initial conditions for the simulation and the lateral boundary conditions for the $64 \times 64\,\mathrm{km}^2$ outer European domain (CD64) are taken from APTA global reanalysis (Sofiev et al., 2018a) and were provided by the Finnish Meteorological Institute (FMI). The global boundary conditions results have been interpolated in time and space to provide hourly boundary conditions for the



outer domain. Boundary conditions for the nested intermediate grid and the two inner grids were calculated on hourly basis from the output of the next-outer grid. For the model simulations with no shipping emissions, the full model chain was run again with all emissions except for those from ship traffic in all the CMAQ grids.

### 2.5 Land-based emissions

Hourly gridded emissions of $NO_X$, sulphur oxides ($SO_X = SO_2 + SO_3$), carbon monoxide (CO), $NH_3$, $PM_{2.5}$, $PM_{coarse}$ and non-methane volatile organic compounds (NMVOC) were calculated for the year 2012 using the comprehensive European emission model SMOKE-EU which is an adaptation of the US-EPA SMOKE (Sparse Matrix Operator Kernel Emissions) model (Bieser et al., 2011a). NMVOC emissions were speciated according to the carbon bond mechanism (cb5) (Yarwood et al., 2005; Passant, 2002) $PM_{2.5}$ emissions according to the AERO5 aerosol mechanism. The SMOKE-EU emission data is

based on reported annual total emissions from the European point source emission register (EPER), the official EMEP emission inventory, and the EDGAR HTAP v2 database (EPER, 2018; CEIP, 2018; Olivier et al., 1999). SMOKE-EU distinguishes 10 major source sectors (including a number of subsector definitions) according to the Selected Nomenclature for sources of Air Pollution (SNAP) of the European Environmental Agency (EEA) (Table 1). For all point sources explicit plume rise calculations based on real world stack information were performed (Bieser et al., 2011b).

The annual total emissions were temporally and spatially redistributed individually for each emission sector and grid cell. Emissions of residential heating were redistributed using the heating demand calculated from daily average temperatures (Aulinger et al., 2011). Emissions from agricultural activity and animal husbandry were disaggregated according to a fertilizer and plant growth model and meteorological parameters (Backes et al., 2016a). Finally, biogenic emissions were calculated off-line with the biogenic Emission Inventory System BEIS version 3.4 (Schwede et al., 2005; Vukovich and Pierce, 2002). The

SMOKE-EU emission datasets were calculated on a $5 \times 5 \, km^2$ grid for the whole of Europe and were subsequently interpolated to the respective CMAQ model grids.

## 3 Shipping emissions and scenario description

### 3.1 Ship emission inventory for the Baltic Sea and North Sea

Shipping emissions for the Baltic Sea and North Sea with high spatial and temporal resolution for this study were obtained

from STEAM (Jalkanen et al., 2009; 2012; Johansson et al., 2013; 2017)). STEAM combines the AIS-based information and the detailed technical knowledge of the world fleet with principles of naval architecture. This input information is used to predict the resistance of vessels in water and the instantaneous engine power of the main and auxiliary engines on a minute by-minute basis, for each vessel that has sent AIS messages. The model predicts as output both the instantaneous fuel consumption and the emissions of selected pollutants. The dynamic modelling of shipping emissions also includes, e.g., the emission control

areas and regulations, emission abatement equipment on-board the ships as well as fuel sulphur content modelling separately for main and auxiliary engines (Johansson et al., 2017; Jalkanen et al., 2012).





Detailed vessel characteristics have been gathered for more than 90,000 individual ships, reported by IHS Fairplay and other ship classification societies. The AIS-system provides automatic updates of the positions and instantaneous speeds of ships at intervals of a few seconds. For this study, archived and down-sampled (approx. 4 minute update rate) AIS messages provided by the Baltic Sea riparian states were used for 2012 and 2014, containing several hundred million AIS messages annually.

The shipping emission inventory consist of hourly updated $2 \times 2\,\mathrm{km}^2$ gridded data for $NO_X$, $SO_X$, CO, and particulate matter, which is further divided into Elementary Carbon (EC), Organic Carbon (OC), sulphate ($SO_4$) and mineral ash. For North Sea, ship emissions from 2011 were adopted for 2012; total ship emissions of $NO_X$ were almost unchanged between the two years. For Baltic Sea ship emissions are from 2012 and were provided for two vertical layers (below 36 m, from 36–1000 m). In CMAQ, $SO_X$ was attributed completely to $SO_2$ and a $NO:NO_2$ ratio of 95:5 was applied. Ship emissions below 36 m were

attributed to the lowest vertical model layer. Ship emissions above 36 m were attributed to the second lowest layer; which appears to be justified based on findings with ship plume simulations (Chosson et al., 2008) showing that plume dispersion in the convective boundary layer (BL) is insensitive to the initial buoyancy flux.

## 3.2 Future scenarios for shipping emissions

Shipping in the Baltic Sea in the future is modelled in a number of scenarios taking into account the development of traffic

and transport work, fleet development for different ship types (number and size), changes in fuel mixture and regulations influencing emissions and fuel consumption. Due to the long lifetime of ships it will take about 30 years after the NECA entry date until the entire ship fleet will be renewed (Kalli et al., 2013) and follows the Tier III emission regulation for $NO_X$. It was decided to perform the future regional CTM simulations for 2040 in order to see the full effect of the NECA.

### 3.2.1 Future baseline scenario "BAU 2040"

The baseline scenario for the future situation in 2040 is the so-called "business as usual" (BAU) scenario that is constructed as a reference scenario ("BAU 2040") forall other future scenarios. It accounts for current trends of economic growth and development of shipping and takes into account already decided regulations. Regarding regulations effecting emissions to air the following are the most important ones in BAU:

1. Sulphur regulation: The Baltic and North Seas are Sulphur Emission Control Areas (SECA) where the maximum allowed
sulphur (S) content in marine fuel has been gradually lowered reaching 0.1 % S from 2015. For sea areas outside SECA the maximum fuel sulphur content will be 0.5 % S from 2020. These regulations directly influence the emissions of $SO_X$ and have a strong impact on the particulate matter emissions.

2. $NO_X$ regulation: $NO_X$ emissions from marine engines are regulated with Tier I for new ships from 2000 and Tier II from
2011. Tier III is applied in $NO_X$ Emission Control Areas and is applied for new ships in the Baltic and North Seas from
    2021.



3. Fuel efficiency: The regulation by IMO on Energy Efficiency Design Index (EEDI) (IMO, 2018) requires new ships to become gradually more fuel efficient. The EEDI regulation was enforced for new ships from 2015 onwards. The EEDI will influence engine emissions in a similar way as the regulations on sulphur and $NO_X$.

The BAU scenario assumes a share of ships driven by liquefied natural gas (LNG) of about 10 % in the ship fleet in 2040. This is modelled as a fraction of new ships introduced each year that will use LNG since retrofitting of existing ships from fuel oil to LNG is assumed less likely due to high costs. Since LNG is used as a means to comply with the sulphur regulations ship types that operate mainly within SECAs are modelled as more likely to use LNG. The fuel efficiency for new ships in BAU is assumed to improve further than what is required from the EEDI regulation, following recent trends and assumption from Kalli et al. (2013), assuming that further technical improvements and more efficient operation take place. The traffic volumes are expected to continue to grow with about 1 % p.a. on average (it varies with ship type); the current trend of using larger vessels is expected to continue as well.

### 3.2.2 Future scenario "NoNECA 2040"

The other two future scenarios, "NoNECA 2040" and "EEDI 2040", are deviations from the development given by the BAU scenario. In the NoNECA scenario, the nitrogen emission control area is assumed not to be implemented, i.e. all new ships up to 2040 are assumed to follow the Tier II $NO_X$ standard. The difference to the BAU scenario is then that new ships from 2021 follows the Tier II standard rather than Tier III. The same introduction of LNG as in BAU is assumed since the use of LNG is mainly motivated by the SECA regulation. From the difference between BAU and NoNECA the effect on emissions of implanting the NECA can be deduced.

### 3.2.3 Future scenario "EEDI 2040"

In the EEDI scenario, improvements in fuel efficiency follow strictly the requirements of the EEDI regulation. Annual efficiency increases of 0.65 % to 1.04 %, depending on ship type, are assumed in the EEDI scenario while the corresponding values in the BAU scenario are 1.3 % to 2.25 %. From the difference between BAU and EEDI the effect of a lower fuel efficiency increase than expected based on continuation of the current trend can be deduced.

Table 2 provides emission scaling factors used in the three scenarios for future shipping emissions.

### 3.3 Future land based emissions

The three scenarios studied here (BAU, NoNECA and EEDI) for future shipping emissions are combined with land-based emissions for 2040 which follow the currently decided emission regulations in Europe. The future land based emission dataset for the year 2040 was created based on the present-day SMOKE-EU emission dataset (Sect. 2.5) using growth factors for each source sector and each species. The employed emission scaling factors are based on the trend between annual total emissions from the 2012 SMOKE-EU inventory and 2040 Baseline emissions of the Current Legislation (CLE) scenario from ECLIPSE v5 (Amann et al., 2014). CLE assumes efficient enforcement of committed legislation but delays in introducing or



enforcing particular laws are considered when such information was available. The scaling factors for land-based emissions, given as average of the Baltic Sea riparian states for CO, PM-other, $SO_4$, $SO_2$, $NO_X$ and $NH_3$ are 0.75, 0.70, 0.45, 0.45, 0.40 and 0.80, respectively.

Ship emissions from the STEAM database were merged with the land-based emissions from the SMOKE-EU database for

the Baltic Sea region and interpolated to the corresponding CMAQ domain sizes and resolutions. Total annual emissions of $NO_X$ in 2012 and in 2040 (BAU scenario) prepared for the CMAQ simulations are shown on geographic maps in Fig. 2.

## 4 Present-day model results

### 4.1 Present-day nitrogen deposition

#### 4.1.1 Comparison of the modelled precipitation with observations

Atmospheric deposition of nitrogen to the Baltic Sea seawater is mainly controlled by wet deposition (Hertel et al., 2003). Since wet deposition of N-containing compounds is determined as the product of the concentration of N-containing compounds dissolved in rainwater and the amount of rainfall, the accurate prediction of the amount, frequency and spatial distribution of precipitation is important. The precipitation amount and frequency from COSMO-CLM output is compared to daily precipitation measurements from rain gauge stations operated by the Swedish Meteorological and Hydrological Institute

(SMHI). 1804 precipitation stations in Sweden were recording daily precipitation sums during 2012; each measurement starting at 6 a.m. UTC on the previous day until 6 a.m. UTC of the current day and is available from the SMHI opendata portal (http://opendata-catalog.smhi.se/explore/).

The station network for daily precipitation collection densely covers the regions of south and middle Sweden (Götaland and Svealand) where each station represents an area of about $10 \times 10\,\text{km}^2$, with lower density of stations in the northern part of

20 Sweden (Södra Norrland and Norra Norrland). Precipitation measurements are known to suffer from evaporation and wind loss, especially on days with very low rain rates (Nespor and Sevruk, 1998). Therefore, a precipitation threshold of $1\,\text{mm}\,\text{d}^{-1}$ is commonly used for the definition of dry days (WMO, 2011). In the present comparison of modelled with observed precipitation no observational threshold was applied in the data analysis. However, in order to determine the performance of COSMO-CLM with respect to the prediction of the number of days without rainfall and hence no wet deposition is examined based on the

25 definition of dry days by a threshold of $0.1\,\text{mm}\,\text{d}^{-1}$ for the model and observational data.

Note for the comparison of modelled and observed precipitation, that the model data is a grid average (either of grid boxes with 0.11 degree or 0.025 degree cell width) whereas the rain gauge data represent a point measurement. Hence, the high-resolution output of COSMO-CLM should better capture the variability of the precipitation measurements. The model-observation comparison was done for the three different configurations of COSMO-CLM: 0.11 degree grid resolution with

30 Tiedtke scheme for convection ("011"), 0.025 degree grid resolution with Tiedtke scheme for convection ("0025_Tiedtke"), and 0.025 degree grid resolution with convection-permitting configuration ("0025_convper"). Fig. 3 shows the monthly precipitation amounts from the three model configurations of the summer months for the Baltic Sea and North Sea region and



compares to the measured precipitation amount at the Swedish rain gauge stations (circles filled with colour indicating the observed value).

Finer grid resolution ("0025_Tiedtke" versus "011") has a tendency to increase the rainfall over land in summer. In particular, more orographic rainfall occurs in Norway for "0025_Tiedtke" compared to "011". The finer resolution improves the agreement with measured rainfall in Svealand in August, but causes too high simulated precipitation in Norrland. The convection-permitting configuration ("0025_convper") yields only small changes compared to "0025_Tiedtke". Most notable differences are the higher precipitation amounts over the Danish islands in June and more convective rainfall over southern Norway in July and August. It has been suggested that the observed inland precipitation intensity in the warm season in the southern part of Sweden is associated with convective rainfall forced by solar heating (Jeong et al., 2011). The slightly increased inland precipitation in June in "0025_convper" compared to "0025_Tiedtke" is in line with this suggestion. However, COSMO-CLM predicts too low precipitation amounts in southern Sweden in June in all three configurations.

Fig. 4 shows the probability distributions of the differences in seasonal averaged (winter months and summer months of 2012) daily precipitation sums from the three model outputs and the observation data in the four regions of Sweden. The percentage fraction of days with zero difference between model and observation and days with difference of $0.1\,\mathrm{mm\,d^{-1}}$ ("delta01" days), corresponding to the threshold value, was calculated. Large deviations between model and observations ($> 10\,\mathrm{mm\,d^{-1}}$) are rare for all three configurations. For "0025_convper", the percentage fraction of days with model-observation deviations below the threshold ("delta01" days) is in the range of 18–34 % in winter and 29–38 % in summer, depending on the region. Compared to the two other configurations, "0025_convper" has the highest percentage fraction of zero difference days both in 2012 and in summer 2012, except for Norra Norrland. The convection-permitting configuration performs better in particular during winter in Götaland, Svealand and S. Norrland, reducing the observation-model difference for too wet days.

The fraction of observed dry days (daily sum $< 0.1\,\mathrm{mm\,d^{-1}}$), as average of all stations of one region, of summer and winter (Table S1) is always higher than the fraction of "delta01" days in the corresponding model data. This implies that COSMO-CLM, on a statistical average, predicts more precipitation days than observed. For "0025_convper", the number of precipitation days, depending on the region, is 9–18 % higher in summer and 9–11 % higher in winter than observed (assuming that the "delta01" days correspond to dry days).

By summing up the model-observation differences of all days in summer and winter averaged over all stations of one region, the model bias for precipitation amounts was determined (Table 3). Note that in this study winter is defined as JFD, including the months January, February and December from 2012 since the simulation was only done for one year. The model tends to predict too dry weather in summer (negative bias for all three configurations) in the southern part of Sweden (Götaland and Svealand). The opposite is the case for the northern part of Sweden (Norrland), where COSMO-CLM has a positive bias. In summer, precipitation from "0025_convper" has a relative bias of -25 %, -19 %, 51 %, and 42 % in Götaland, Svealand, S. Norland, N. Norland, respectively, compared to observed precipitation amounts. In winter, precipitation from "0025_convper" has a much lower relative bias (-4 %, -7 %, 3 %, and 32 % in Götaland, Svealand, S. Norland, N. Norland, respectively). A possible reason for the dry bias in summer could be that south Sweden receives too little precipitation due to its location in the lee of the Norwegian mountains, where humidity is lost through excessive orographic rainfall in the simulation.



The probability distribution of differences for all months for Götaland is shown in the Supplement (Fig. S1). The convection-permitting configuration "0025_convper" reduces the frequency of negative differences in the range of 0.1–4 mm d$^{-1}$ for daily rainfall in June compared to "0025_Tiedtke". The frequencies of negative differences between observation and model is similar for all summer months, while observed total monthly precipitation amount is highest in June (Götaland station average, June: 111 mm, July: 87 mm, August: 68 mm). COSMO-CLM performs better in the cold season (October - March) in Götaland when differences between observations and model are in the range between -4 and +6 mm d$^{-1}$ for more than 90 % of the time (Fig. S1).

### 4.1.2 Evaluation of the modelled wet deposition of nitrogen

Wet deposition of oxidised and reduced nitrogen was evaluated with measurements of regional background stations in the Baltic Sea region for the period of 1 March to 30 November 2012. The winter months were excluded from the analysis to avoid possible artefacts associated with the collection of snow. Modelled wet deposition of nitrate, $NO_3^-$ (WNO$_3$), representing oxidised nitrogen and modelled wet deposition of ammonium, $NH_4^+$ (WNH$_4$), representing reduced nitrogen, were compared to data from the EMEP monitoring programme (Tørseth et al., 2012; EMEP, 2014) at the stations displayed on the map in Fig. 5a. Observation data was obtained from the EBAS database (http://ebas.nilu.no/). Modelled wet deposition of HNO$_3$ was included in WNO$_3$ and modelled wet deposition of NH$_3$ was included in WNH$_4$, because it is assumed that the gases are partially or fully dissolved in the sampled rainwater. The summation also gives a more robust estimate of the wet deposition of oxidised and reduced nitrogen. Measured concentrations of $NO_3^-$ and $NH_4^+$ in rainwater were converted into nitrogen deposition per area by the amount of rainwater measured at the respective station. Daily sums of wet deposition were calculated from the volume-weighted concentrations multiplied by the precipitation amounts. The comparison of the daily sum of wet deposition is done in terms of mean values ($\mu_{\mathrm{Mod}}$ and $\mu_{\mathrm{Obs}}$), the Spearman's correlation coefficient ($R_{\mathrm{Spr}}$) and the normalized mean bias (NMB). Only days with predicted and observed rain events in common were included in the comparison. Several stations in the Baltic Sea region had only few measurements during the period. Stations with less than seven model-observation pairs were excluded from the statistical analysis. CMAQ model data from the intermediate grid (CD16) and from the high-resolution grid (CD04) are evaluated separately.

Plots in Fig. 5b–g show the time series modelled and observed daily sums of WNO$_3$ at selected stations (all other stations are shown in Fig. S2). The 4-km resolution output gave higher WNO$_3$ than the coarser CD16 output in the southern part of the Baltic Sea region (e.g. stations Zingst, Preila and Keldsnor). For the more northern stations, simulated time series of WNO$_3$ from the two model grids are similar. The correlation between modelled and observed data improves for several stations when going from CD16 to CD04, supporting the use of finer resolution for chemistry and transport computations in combination with high-resolution precipitation modelling. WNO$_3$ is underestimated at all stations included in the statistical analysis (Table 4), most severely at the Finnish stations and at Zingst (NMB between -0.75 and -0.90).

WNH$_4$ is underestimated at all stations included in the statistical analysis (Table 5; corresponding time series are plotted in Fig. S3). The underestimation is highest for Zingst and the Finnish stations, as for WNO$_3$. The joint underestimation of WNO$_3$ and WNH$_4$ especially in the northern part of the Baltic Sea region could indicate missing formation of particulate ammonium





nitrate or too slow conversion of $NO_X$ to $HNO_3$ in the model. The long-range transport of particulate ammonium to the remote parts of the Baltic Sea region is further limited by the availability of particulate nitrate and sulphate (Ferm and Hellsten, 2012).

To account for the fact that the days with predicted rain often do not correspond to days with observed rain, seasonal averages (spring, summer and autumn) were calculated for $WNO_3$ (Table S2) and $WNH_4$ (Table S3) independently for CD04 model data and observation data. The joint underestimation of $WNO_3$ and $WNH_4$ at Zingst and the Finnish stations is confirmed in this analysis. At Zingst, seasonal averages of $WNO_3$ based on model data are lower by -52 % to -63 % and seasonal averages of $WNH_4$ based on model data are lower by -64 % to -79 % than the corrsponding seasonal averages based on observations. The ratio of modelled to observed seasonal average of $WNO_3$ shows little variation for the three seasons (0.37–0.48). At Preila, a coastal station in Lithuania, both observed averages of $WNO_3$ and $WNH_4$ are underestimated in spring and autumn, but not in summer. The ratio of modelled to observed seasonal averages of $WNO_3$ and $WNH_4$ show a consistent seasonal pattern at the nine stations (Fig. S4), pointing to the formation and atmospheric transport of particulate ammonium nitrate as common cause.

The agricultural sector, including animal husbandry, is an important source of r educed nitrogen emissions to the atmosphere (e.g., Bouwman et al., 1997). $NH_3$ emissions from animal housing and application of manure on fields are highly relevant and can influence the formation of ammonium nitrate particles (Backes et al., 2016b). Formation of ammonium sulphate is much less sensitive to agricultural $NH_3$ emissions because ambient background concentrations of $NH_3$ in the model simulations are high enough to saturate the reaction forming sulphate particles (Backes et al., 2016b). Too low emissions of gaseous $NH_3$ from agriculture in northern Germany might also explain the missing $WNH_4$ at Zingst. Annual emission totals of $NH_3$ reported by Germany under the Long-range Transboundary Air Pollution (LRTAP) convention over the period 2009–2015 raised by ca. 9 % over prior estimates, mainly due to additional emissions from the use of inorganic and organic fertilizers (EEA, 2018; EEA, 2014). These additional reported emissions had not been included in the SMOKE-EU emission inventory at the time of the model simulations.

Measurements of gaseous $NH_3$ from spring to autumn 2012 were available for the stations Anholt, Tange and Risoe in Denmark and for Diabla Gora in Poland. At all four stations, CMAQ overestimated the observed $NH_3$ concentrations (NMB range 0.40–0.92), indicating that the availability of acidic compounds (such as $HNO_3$ and $H_2SO_4$) rather than that of $NH_3$ limited the formation of particulate ammonium in the southern part of the Baltic Sea region in the simulations.

### 4.1.3 Nitrogen deposition to the Baltic Sea region

Deposition of nitrogen includes particulate ammonium and nitrate as well as gaseous NO, $NO_2$, $NH_3$, nitrate radical ($NO_3$), $HNO_3$, dinitrogen pentoxide ($N_2O_5$), peroxy nitric acid ($HNO_4$) and peroxy acetyl nitrate (PAN). Figure 6a shows the spatial distribution of the annual total (wet and dry) nitrogen deposition in 2012 from the CMAQ simulation. A strong gradient from southwest to northeast is found for the annual total nitrogen deposition, both over land and over sea. Highest nitrogen deposition (range 500–650 mg(N) m$^{-2}$) to seawater is found for Belt/Kattegat and Arkona Basin areas. Seasonally accumulated nitrogen deposition to the Baltic Sea seawater shows low values (below 90 mg(N) m$^{-2}$) in winter and spring and higher values (70–270 mg(N) m$^{-2}$) in summer and autumn (Fig. S5). From spring to autumn there is a clear gradient between land and sea, with 2–3 times higher nitrogen deposition over land, which relates to the canopy uptake by vegetation. In winter months, the picture



changes and land and sea receive similar amounts of nitrogen deposition. Over the Baltic Sea, highest nitrogen deposition is predicted for the autumn months (SON), with maximum values of 230 mg(N) m$^{-2}$ in the northern Baltic Proper.

In coastal regions, nitrogen deposition is markedly higher compared to further inland. Sea-salt particles can considerably increase nitrogen deposition in coastal regions, although this effect is relatively small in the Baltic Sea region and only pro-
nounced along the coast of Denmark (Neumann et al., 2016a). Reaction of HNO$_3$ with coarse mode sea-salt particles, when marine aerosol mixes with the polluted air from the continent, leads to a shift of fine mode nitrate to the coarse mode, through the formation of sodium nitrate (Brimblecombe and Clegg, 1988; Zhuang et al., 1999) which is essentially non-volatile in atmospheric conditions. Since coarse mode particles are prone to deposition through gravitational settling, the nitrate formation reaction on sea-salt particles may lead to enhanced deposition of nitrogen in the coastal zone (Spokes et al., 2000; Neumann
et al., 2016a).

The injection of reactive nitrogen through shipping activities contributes to increased input of nitrogen to the Baltic Sea. The annual nitrogen deposition related to ship emissions (ship-related deposition) is on average 52 mg(N) m$^{-2}$ over the Baltic Sea (Fig. 6b). The absolute contribution of shipping emissions (seasonal cycle shown in Fig. 6S) is highest during summer; amounting to 20 mg(N) m$^{-2}$ (JJA) in the Baltic Sea on average.
Table 6 summarizes the annual and seasonal sums of reduced, oxidised and total nitrogen deposition amounts to the seawater of the Baltic Sea together with the deposition amounts related to shipping. Total annual nitrogen deposition to Baltic Sea is 29 % lower than the estimate from the EMEP-MSC/W model, normalised by the inter-annual changes in meteorological conditions, used in the HELCOM (Baltic Marine Environment Protection Commission - Helsinki Commission) evaluation of the Baltic Sea marine environmental status (2012: 223.6 kt N y$^{-1}$; Bartnicki et al., 2017). The annual reduced and oxidized nitrogen deposition
is lower by 33 % and 27 %, respectively, than the EMEP data for 2012.

## 4.2 Present-day air quality

CMAQ model results for surface air concentrations of O$_3$, NO$_2$, SO$_2$ and PM$_{2.5}$ from the 4-km resolution grid were evaluated against measurements at regional background stations of the EMEP monitoring programme available from the EBAS database. In addition to the statistical indicators used in the evaluation of deposition, the root mean square error of the modelled values
(RMSE) is included in the evaluation of air concentrations, a frequently used measure of the differences between values predicted by a model and the values actually observed. The evaluation was done for the entire year 2012 and separately for summer (JJA) 2012. In the context of this evaluation of predicted air pollutant concentrations, we consider a correlation coefficient of more than 0.5 to indicate a correlation between modelled and observed time series, while values of 0.7 and above are considered as a good correlation.

### 4.2.1 Seasonality of ozone and comparison with measurements
Ozone is generated in the troposphere involving two classes of precursor compounds, VOC and NO$_X$, in photochemical reaction cycles, initiated by the reaction of the OH radical with organic molecules. O$_3$ is harmful to vegetation, reducing plant primary productivity and agricultural crop yields. O$_3$ is also a serious concern for human health. The precursors of O$_3$ have





anthropogenic and natural (or biogenic) sources, both are considered in the CTM simulation. At the continental scale, the formation of $O_3$ is sustained by the oxidation of methane ($CH_4$) and CO.

In the present-day CMAQ simulation, highest seasonal averages of the daily maximum $O_3$ concentration were found in spring (MAM), with levels up to 50 ppbv in the southern part of the Baltic Sea region (Fig. S7), which are a consequence of the

inflow of ozone-rich background air masses from the Atlantic. Photochemical production in summer leads to elevated ozone concentrations over the southern Baltic Sea (range 36–44 ppbv). In autumn and winter daily maximum $O_3$ concentrations in the Baltic Sea region are below 34 ppbv.

The results for the statistical evaluation of modelled daily mean $O_3$ concentrations are summarized in Table 7. Modelled daily means of $O_3$ are in good agreement with measurements at all stations ($R_{Spr} = 0.75$, RMSE = 6.8 ppbv, both as average

of all stations; NMB range: -0.16 to -0.02) when the entire year is considered. In summer, ozone is slightly underestimated at the stations in the southern part of the Baltic Sea region (NMB range: -0.23 to -0.12). The overall agreement in summer is, however, fairly good ($R_{Spr} = 0.62$, RMSE = 6.9 ppbv; both as average of all stations).

### 4.2.2   Seasonality of nitrogen dioxide and comparison with measurements

The main sources of nitrogen oxides are traffic and combustion processes. Emissions of $NO_X$ and the derived oxidation prod-

ucts strongly influence concentrations of ozone and particulate matter (Seinfeld and Pandis, 2005), the latter directly through formation of nitrate aerosols and indirectly by influencing the oxidation of secondary aerosol precursors.

In spring and summer, average $NO_2$ concentrations in proximity of the main shipping routes several times exceed the concentrations in the regional background (Fig. S8). In autumn and winter the spatial distribution of modelled seasonal averages show a gradient from south to north. High values are predicted in northern Germany, Poland and over the Danish Straits (range:

3.5–7.5 ppbv) with hotspots in the large cities (> 9 ppbv). The wider spread of elevated $NO_2$ concentrations in winter compared to summer is in accordance with a longer lifetime of $NO_X$ in winter (up to one day) compared to summer (a few hours) (Schaub et al., 2007).

The evaluation of modelled $NO_2$ based on daily concentrations for the entire year and for summer (Table 8) indicates a better performance of CMAQ over the entire year than over summer alone. A good correlation is obtained at 9 out of 12 stations for

the entire year, whereas only 5 stations show a good correlation in summer.

In contrast to a previous study with the CMAQ model in the North Sea region by Aulinger et al. (2016) and other multi-model air quality studies in Europe (e.g., Giordano et al., 2015), the simulations for the Baltic Sea region did not show substantial underestimation of observed $NO_2$ daily means; NMB is positive at 11 stations for the entire year and at 8 stations for summer. The CD04 simulations rather predict slightly higher $NO_2$ concentrations than observed at most stations (NMB range: -0.28–

0.44 for the year; NMB range: -0.31–0.83 for JJA; average of all stations). RMSE is in the range 1.0–3.2 ppbv for the year and in the range 0.2–3.5 ppbv for summer. The improved performance for $NO_2$ compared to the previous study by Aulinger et al. (2016) is partly attributed to the high spatial resolution, as $NO_X$ emissions are injected into a smaller grid box volume and consequently less diluted initially.



### 4.2.3 Seasonality of sulphur dioxide and comparison with measurements

The main atmospheric sources of $SO_2$ are fossil fuel combustion and metal producing industries. The atmospheric lifetime of $SO_2$ based on the reaction with the OH radical is about one week (Seinfeld and Pandis, 2005). $SO_2$ is removed efficiently by dry deposition; the lifetime towards dry deposition is typically about one day. Overall, the average lifetime of $SO_2$ in the troposphere is a few days. $SO_2$ is converted to sulphate aerosols either via gas-phase oxidation to $H_2SO_4$ and subsequent nucleation or condensation or by uptake into cloud droplets followed by aqueous phase oxidation. $SO_2$ is a major air pollutant and linked to air quality and human health issues.

$SO_2$ shows higher concentrations in autumn and winter than in spring and summer (Fig. S9). The main reason is the stable boundary layer connected with stagnant air and frequent inversions during the colder season which causes emissions of $SO_2$ to accumulate in the surface layer. Residential heating emissions and power plant emissions for district heating strongly contribute to the higher $SO_2$ concentrations in winter as compared to summer. Highest $SO_2$ concentrations in autumn and winter are simulated over Poland, where levels in the cities exceed 3 ppbv. In spring and summer elevated $SO_2$ levels over the Baltic Sea (0.9–1.8 ppbv), confined to the main shipping routes, are a sign of the influence from shipping activities. Another factor leading to lower concentrations in summer is the faster oxidation of $SO_2$ by OH compared to other seasons.

Simulated $SO_2$ daily mean concentrations are correlated with the observed daily mean concentrations at all stations of the Baltic Sea region for the entire year but in summer they are not correlated at several stations (Table 9). The associated RMSE is relatively high (RMSE = 1.01 ppbv for the year, RMSE = 0.41 ppbv for JJA; average of all stations) but the summer RMSE is lower than in the multi-model study by Giordano et al. (2015) (RMSE range: 2.17–2.34 ppbv). Observed $SO_2$ concentrations are generally overestimated (NMB range: -0.04–1.62 for the year; NMB range: -0.07–1.84 for JJA). In particular, summer mean $SO_2$ at the remote stations Ähtäri II (Finland) and Preila (Lithuania) is overestimated by a factor of 2–3, indicating that the oxidation of $SO_2$ in the background air is not efficient enough in the simulation. The overestimation of both $SO_2$ and $NO_2$ by the model corroborates the hypothesis of too slow conversion of the primary gaseous precursors given in Sect. 4.1.2 to explain the underestimated nitrogen deposition, but it is also possible that the anthropogenic emissions of these pollutants are too high in the model.

### 4.2.4 Seasonality of PM$_{2.5}$ and comparison with measurements

Particulate matter (PM) is a wide-spread air pollutant, consisting of a mixture of solid and liquid particles suspended in air. Ambient $PM_{2.5}$ comprises primary emitted and secondary PM that formed in the atmosphere. Primary PM includes OC and EC particles from anthropogenic sources such as traffic and industrial activities, as well as wind-blown soil dust and sea-salt particles from natural sources. Secondary PM includes secondary inorganic and organic particles from the homogeneous and heterogeneous chemical transformation of primary gaseous precursors such as $NO_X$, $SO_2$, $NH_3$ and NMVOC in the atmosphere. PM between 0.1 μm and 1 μm in diameter can remain in the atmosphere for days or weeks and thus be subject to long-range transport. $PM_{2.5}$ is known to have adverse health effects; short-term exposure to $PM_{2.5}$ is associated with respiratory and





cardiovascular diseases (e.g., Pope and Dockery, 2006), while long-term exposure to $PM_{2.5}$ is associated with an increase in the long-term risk of cardiopulmonary mortality (Beelen et al., 2008).

Modelled $PM_{2.5}$ is highest in winter, exceeding $6\,\mu g\,m^{-3}$ in most parts of the Baltic Sea region, which is attributable to the stagnant conditions and higher emissions of primary PM than in the other seasons (Fig. S10). Low temperatures in winter are
favourable for the condensation of gaseous precursors to particles. In spring and autumn, $PM_{2.5}$ is higher in the southern part, both over land and sea, than in the northern part of the Baltic Sea region. The high $PM_{2.5}$ levels over land in the south are presumably due to a combination of land-based PM emissions, long-range transported PM and the condensation of secondary PM from the transformation of gaseous precursor emissions. In summer, $PM_{2.5}$ in the region is much smaller and shipping activities influence $PM_{2.5}$ levels over the Baltic Sea, as indicated by elevated concentrations along the shipping routes in the
Danish Straits and the Gulf of Finland.

Data from eight stations was available for the comparison of modelled against observed $PM_{2.5}$ (Table 10). For the entire year CMAQ performs quite well in the prediction of daily mean $PM_{2.5}$ ($R_{Spr} = 0.57$, NMB = -0.22, RMSE = $5.6\,\mu g\,m^{-3}$; each as average of all stations). In the summer period, $PM_{2.5}$ is underestimated at all stations (average NMB = -0.60). This is partly due to the underestimation of secondary organic aerosols by the CMAQ model. Although the capability of CMAQ to predict SOA
has been improved compared to earlier versions of the model, the predicted SOA compounds make up only a small fraction of the predicted $PM_{2.5}$. On the other hand, the contribution of SOA is relatively small at coastal sites (about $0.1\,\mu g\,m^{-3}$) compared to inland sites (about $0.5\,\mu g\,m^{-3}$) in northern Europe (Andersson-Sköld et al., 2001; Gelencsér et al., 2007). Other causes for the low $PM_{2.5}$ concentrations in summer could be too little formation of SIA due to the inefficient conversion of primary gaseous precursors, as stated in Sect. 4.2.3. In addition, emissions of wind-blown soil dust particles were not activated in the CMAQ
simulations. A deeper investigation of the reasons for the underestimation in summer would require a detailed comparison of the individual aerosol components, which is out of the scope of the present study.

### 4.2.5  Summer mean ship contribution of air pollutants

The influence of shipping emissions on the present-day air quality was evaluated for the summer months. The results for the impact of shipping emissions were calculated as difference between the reference run and the run with no ship emissions (in
the North and Baltic seas) in 2012. Results for the absolute and relative ship contributions in summer (as JJA average) are shown in Fig. 7 for the daily maximum $O_3$, $NO_2$, $SO_2$ as well as $PM_{2.5}$, and discussed in the following.

In the proximity of the main shipping routes, ozone concentrations are reduced by 10–20 % on spatial average in summer compared to a situation with no shipping emissions. This reduction is due to local scale titration of $O_3$ by NO emitted in the ship plumes. With increased distance (> 100 km) from the main ship routes, photochemical ozone production takes place when
$NO_X$ and CO from ship exhaust mixes with the continental emissions of NMVOC. Shipping emissions contribute to summer daily maximum $O_3$ in the coastal areas of the Baltic states, southern Finland and eastern Sweden by up to 4.5 ppbv (ca. 20 %) (Fig. 7a). A limitation of the model results for regional surface concentrations of $O_3$ over the Baltic Sea region is the lack of emission data on NMVOC from shipping in the STEAM inventory. Additional NMVOC emissions from shipping would enhance photochemical ozone production.





Summer mean surface air concentrations of $NO_2$ over the Baltic Sea in the background areas without shipping are up to 3.5 ppbv, while along the main shipping routes concentrations of up to 8 ppbv are reached (Fig. 7b). $NO_2$ decreases to background values within a few hundred kilometres distance from the centre of the shipping routes. From the model simulations it is evident that shipping emissions are the main contributor to ambient $NO_2$ concentrations over the Baltic Sea in summer.

Ships emit $NO_X$ mainly in the form of nitrogen oxide (NO). When ozone entrains into the ship's exhaust plume, NO is however quickly converted to $NO_2$, so atmospheric $NO_X$ will be mainly in the form of $NO_2$.

Over the Baltic Sea, shipping emissions have a high contribution to atmospheric $SO_2$ concentrations in the present-day situation. The summer mean ship contribution to $SO_2$ is 2.5 ppbv (about 80 %) or more in a wide area around the main shipping routes of the Baltic Sea (Fig. 7c). The EU has implemented a sulphur emission control area (SECA) for the North and Baltic

seas, which means that in the present-day situation for the model (year 2012), fuels burned on ships in these areas must not contain more than 1.0 % S. After 1 January 2015, not more than 0.1 % S in the fuel is allowed in the SECA, which drastically decreases $SO_2$ concentrations along the shipping routes (Kattner et al., 2015).

The ship contribution to summer mean $PM_{2.5}$ shows a gradient from south to north with highest concentrations over the Belt Sea/Kattegat and over the sea south of Sweden with maximum values up to $1.4\,\mu g\,m^{-3}$ (Fig. 7d). The ship contribution

is highest along (up to 50 %) the main shipping routes between Denmark and St. Petersburg. Over land, the relative ship contribution is below 30 %. The relative ship contribution in the coastal regions tends to be overestimated by the model due to the underestimation of ambient $PM_{2.5}$ in summer (Sect. 4.2.4). The influence of ship emissions on $PM_{2.5}$ extends over a wider corridor over the Baltic Sea than this is the case for $NO_2$ and $SO_2$. This can be attributed to the formation of secondary particles in the ship exhaust plume during its transport away from the shipping route. The production of secondary particles

via the oxidation of $NO_2$ and $SO_2$ emitted from ships happens over a longer time scale, during which the plume is advected. In addition, the aerosol formation rates critically depend on ambient temperature, humidity, solar radiation and the level of atmospheric oxidants (OH and $NO_3$ radicals) and reaction partners such as $NH_3$.

## 5   Future scenario model results

### 5.1   Air quality changes in 2040 compared to present-day

#### 5.1.1   Future air quality situation

In the "BAU 2040" scenario (future reference simulation), with the introduction of the NECA in 2021, $NO_X$ emissions from ship traffic in the Baltic Sea are reduced by 79 % in 2040 compared to 2012, because most ships of the Baltic Sea ship fleet will then fulfil the Tier III regulation. In the NoNECA scenario, the NECA is not established, but all other developments (economic growth, fleet renewal and efficiency increase) are as in the BAU scenario, still leading to a reduction of $NO_X$ emission from

30 ships by 50 %. In the EEDI scenario, fuel efficiency increase follows the EEDI regulation, thus remaining below the efficiency increase assumed for the BAU scenario, resulting in an overall reduction of $NO_X$ emissions from ships by 71 % compared to



2012. The spatial maps of average summer (JJA) concentrations of daily maximum $O_3$, $NO_2$, $SO_2$, and $PM_{2.5}$ in the three future scenarios for 2040 are compared to the present-day results in Fig. 8.

Over most parts of the Baltic Sea region, the summer mean of daily maximum $O_3$ in "BAU 2040" decreases by 10–25 % compared to 2012, as consequence of the NECA and reduced land-based emissions of $NO_X$ (Fig. 8a). The future change of

ozone is similar in "EEDI 2040", implying, that the effect of increased fuel efficiency is less pronounced and that the $NO_X$ reduction through establishing the NECA has a much greater influence on future ozone levels in the Baltic Sea region. In the NoNECA scenario, daily maximum $O_3$ over land will decrease less than in the BAU scenario, but still an average ozone reduction by 15 % in 2040 is predicted for large parts of Sweden and the Baltic Sea, compared to present day.

In the "BAU 2040" scenario, summer mean $NO_2$ concentrations are drastically reduced, by ~80 % over most parts of the

Baltic Sea and by up to ca. 90 % in the northern Baltic Proper, compared to 2012 (Fig. 8b). This appears to be a result of the combined emission reductions through the NECA and the regulation of land-based emissions (Sect. 3.3), leading to a shift in the overall atmospheric photochemical regime due to the lower abundance of $NO_X$ in the future. Strong reduction is also seen in "EEDI 2040", where $NO_2$ levels over the Baltic Sea decrease by ~80 %, compared to 2012. "NoNECA 2040" results in a reduction of $NO_2$ by ~50 % over the entire Baltic Sea.

"BAU 2040" adopts the agreed $SO_X$ emission reduction measures; i.e. the SECA limit of 0.1 % S in fuel from 2015 onwards and the global limit of 0.5 % S in fuel from 2020 onwards. The other two future scenarios also implement the two sulphur regulations. In 2040, summer mean $SO_2$ levels drop by 80–90 % over the entire Baltic Sea compared to present day.

Summer mean $PM_{2.5}$ levels in 2040 decrease by 50–60 % along the main shipping routes and by 40-50 % in the other parts of the Baltic Sea, compared to 2012. The EEDI scenario involves lower primary PM emission reductions (by 51 %) than in

"BAU 2040" and "NoNECA 2040" (by 65 %). However, as for the other air pollutants, no large differences of the spatial concentration distributions in summer 2040 are seen between the EEDI and the BAU scenarios, suggesting that the lower fuel efficiency increase has only marginal implications on the future air quality in the Baltic Sea region.

### 5.1.2   Influence of ship emissions in the BAU future scenario

Figure 9 summarizes the predicted ship contribution in summer 2040 according to the "BAU 2040" scenario, analogous to

Fig. 7 for the present-day ship contribution. As a result of the introduction of the NECA in 2021, the future impact of ship emissions on $O_3$ levels in the Baltic Sea region diminishes. In 2040, the ship contribution to summer mean daily maximum $O_3$ concentrations is highest over the Gotland Basin (range: 5–6 ppbv), while it is smaller for all over parts of the Baltic Sea region, not exceeding 4.5 ppbv. Overall, the model simulations predict that shipping emissions will still influence ozone levels over the Baltic Sea and in the coastal areas in 2040, with relative contributions in the range of 10–20 % to daily maximum $O_3$.

The absolute ship contribution to summer mean $NO_2$ concentrations in 2040 drop substantially compared to 2012. The ship-related $NO_2$ concentration decreases from ca. 3 ppbv in the present-day situation to 0.5–1.5 ppbv in the BAU scenario, along the main shipping routes. Even with the NECA established, emissions from ship traffic remain the dominant contributor to atmospheric $NO_2$ over the Baltic Sea in 2040.



The absolute ship contribution to $SO_2$ concentrations in summer 2040 is less than 0.1 ppbv. However, the ship influence on ambient $SO_2$ concentrations has not completely vanished in 2040. Along the main shipping routes throughout the Baltic Sea, the relative contribution remains high.

The absolute ship contribution to $PM_{2.5}$ in summer 2040 is predicted to be $\leq 0.2\,\mu g\,m^{-3}$ over most parts of the Baltic Sea region, with higher values over the Belt/Kattegat ($0.4\,\mu g\,m^{-3}$). The ship influence substantially weakens compared to the present-day situation: the relative contribution peaks along the shipping routes (15–25 %) and is below 10 % over land.

### 5.1.3  Future change of the ship contribution

Figure 10 shows the future change of the ship contribution in summer 2040 compared to 2012, when following the "BAU 2040" scenario. Future changes of the ship contribution to daily maximum $O_3$ are divided into two regions with opposing sign, one with a relative increase, over the central shipping routes, and one with a relative decrease, outside the ship tracks and over the coastal regions. Over the ship lanes, ozone recovers due to reduced titration of ozone in the ship plumes following the lower emissions of NO from ships. In greater distance from the ship lanes, photochemical production of ozone declines compared to present day, giving raise to lower $O_3$ concentrations.

The ship contribution to $NO_2$ decreases by 80–85 % over the Baltic Sea, slightly more than linear with the reduced $NO_X$ emissions from shipping. The decrease is smaller (~77 %) in some port cities like Gdansk and St. Petersburg and in areas with high density of ship traffic. The reduced $NO_X$ emission from ships causes an increase of the ratio of $[NO_2]$ to $[NO]$ (short: $NO_2$-to-NO ratio) in the ship plumes. Although the $NO_2$-to-NO ratio at the ship stack is the same (equal to 5:95), it becomes higher, as $NO_2$ from the background air entrains into the plume, than in the present-day situation. According to the photostationary state relation, the increased ratio causes a higher steady-state $O_3$ concentration in the ship plume. With the local increase of $O_3$, the reaction of NO with the hydroperoxyl ($HO_2$) radical giving $NO_2$ starts to compete with the titration reaction (reaction of NO with $O_3$). In the reaction of NO with $HO_2$ an additional ozone molecule is produced, as the resulting $NO_2$ molecule photolyses, amplifying the ozone production in the plume. Hence the smaller decrease of the $NO_2$ ship contribution is due a change of the photochemistry regime in the ship plumes accompanied with a higher conversion of NO to $NO_2$.

For the ship contribution to $SO_2$, a uniform decline by around 90 % is seen for the entire Baltic Sea, in accordance with a linear decrease following the reduction of $SO_X$ emissions from shipping by 91.2 % between 2012 and 2040 in "BAU 2040". Note that ship emissions of $SO_X$ were attributed completely to $SO_2$. As for the $NO_2$ ship contribution, the decrease is slightly higher than expected due to the reduction of ship emissions. Due to the drastic decrease of nitrogen oxides, the atmospheric oxidation capacity increases in the future scenario simulation leading to more efficient oxidation of pollutants and higher availability of photo-oxidants (OH and $HO_2$ radicals). Hence, the removal rates of $SO_2$ and $NO_2$ by reaction with photo-oxidants and the rate of $SO_2$ oxidation in clouds are slightly increased in 2040 compared to 2012.

The ship contributed summer mean $PM_{2.5}$ between 2012 and 2040 ("BAU 2040") reduces by 75–90 %, with largest reductions over the southern part of the Baltic Sea and in the coastal regions. This is more than can be explained by the reduction of primary PM emissions (by 65 %) from shipping. Thus a substantial fraction of the changed ship contribution is caused by



changes of the secondary aerosol production. The future ship contribution to $PM_{2.5}$ is affected by reduced $SO_X$ emissions from ships, as a result of the regulations for lower sulphur fuel content and by reduced $NO_X$ emissions due to the NECA.

Together, the regulations lead to a decline of the atmospheric formation of sulphate and nitrate particles related to shipping. In the southern part of the Baltic Sea region, especially over Denmark and northern Germany, the ship-related formation of sec-

ondary aerosol is also affected by the lower $NH_3$ emissions from agriculture. Decreasing atmospheric ammonia concentrations reduces the formation of ammonium nitrate particles, since their formation is limited by the availability of $NH_3$.

For the other two future scenarios, "NoNECA 2040" and "EEDI 2040", changes of the ship contributed pollutant concentrations compared to present day are smaller than in "BAU 2040". In the scenario without implementation of NECA, "NoNECA 2040", the ship contribution to $NO_2$ in 2040 decreases by 50–60 % over the Baltic Sea (Fig. S11). The ship contribution to

ozone increases widely by more than 10 % compared to present-day, indicating enhanced ozone production due to shipping activities in 2040, mainly over sea and the coastal areas of Sweden, Denmark and Poland. The EEDI scenario, with lower fuel efficiency, results in a significantly smaller reduction of ship contributed $PM_{2.5}$ than the BAU scenario. Still, the ship contributed summer mean $PM_{2.5}$ between 2012 and 2040 reduces by 65–80 % over the impacted areas (Fig. S12).

## 5.2 Future air quality: effect of the NECA

The difference in the two future scenarios "BAU 2040" and "NoNECA 2040" is the higher emission reduction of $NO_X$ from shipping in the BAU scenario through establishment of the NECA. Figure 11 illustrates the effect of introducing the NECA in 2021 on major air quality components compared to a future situation without NECA, determined based on the difference between modelled concentrations in the "BAU 2040" and "NoNECA 2040" scenarios. Land-based emissions are the same in both scenarios, therefore changes are solely due to different ship emissions in the two future scenarios.

The result of the NECA in 2040 is a reduction of $NO_X$ emissions from shipping by 59 % on average, corresponding to the difference between a Tier III dominated ship fleet with the NECA and Tier II dominated ship fleet without the NECA. The reduction of $NO_X$ emissions from shipping primarily translates into a ~60 % decrease of $NO_2$ summer mean concentrations within a wide corridor of the ship routes. In addition, the population in coastal areas in northern Germany, Denmark and western Sweden will be less exposed to $NO_2$ in 2040 due to the introduction of the NECA. Due to the lower atmospheric $NO_X$

levels, less ozone is formed, and daily maximum $O_3$ concentration over the Baltic Sea in summer 2040 is on average 6 % lower than without the NECA. In the areas close to the main shipping routes, ozone is almost unchanged despite the sharp reduction of $NO_X$ emissions, probably due to compensating effects between changed titration losses and changed photochemical ozone production. As expected, levels of atmospheric $SO_2$ are largely unaffected by the NECA ($< \pm 2$ %).

A secondary effect of the NECA is a reduction of the formation of particulate nitrate. Due to the non-linearity of the

atmospheric particle mass formation, i.e. photochemistry and gas-to-particle conversion depend on precursor concentrations and existing particulate matter in a non-linear fashion, the impact of reducing gaseous precursors does not result in a linear reduction of future $PM_{2.5}$ levels. Fig. 11d shows the change of summer mean $PM_{2.5}$ concentration pattern due to the NECA. Note that primary emissions of $PM_{2.5}$ are the same in BAU and NoNECA, thus changes are solely attributed to modified



particulate nitrate concentrations. Largest decrease of PM$_{2.5}$, by up to 8 %, occurs over the Danish islands, where the abundance of ammonium nitrate is highest.

## 5.3 Future air quality: effect of lower fuel efficiency

The BAU scenario assumes an improvement of the marine fuel efficiency beyond that required by the EEDI regulation for new ships. With the difference between the "EEDI 2040" and "BAU 2040" scenarios (land-based emissions are the same in both scenarios), the effect of a slower rate of fuel efficiency improvement compared to the projections in the BAU scenario on the air quality in 2040 is determined. The lower fuel efficiency affects the ship engine emissions and leads to NO$_X$, SO$_2$ and PM$_{2.5}$ emissions from ships that are on average 37.9 %, 36.8 % and 39.6 % higher in 2040, respectively, compared to the BAU scenario. As a consequence of the lower fuel efficiency, modelled summer mean concentrations of NO$_2$ and SO$_2$ along the main shipping routes in 2040 are higher by 40 % and 25 % than in BAU, respectively (Fig. 12).

The lower fuel efficiency has little influence on daily maximum ozone concentrations over the Baltic Sea. Further, the influence of the changed fuel efficiency on atmospheric secondary particle formation is rather limited (not shown). For PM$_{2.5}$, the higher primary particle emissions compared to BAU do not fully propagate into surface air concentrations (increase by less than 10 %). A large fraction of the ship-related PM$_{2.5}$ is from secondary formation, which does not increase proportionally with the increased primary PM emissions, for example due to the limited availability of NH$_3$.

## 5.4 Future nitrogen deposition

Summer-accumulated total nitrogen deposition to seawater in 2040 according to "BAU 2040" is below 100 mg(N) m$^{-2}$ in most parts of the Baltic Sea, with highest deposition remaining in the Belt Sea (Fig. 13a). The average summer deposition rate to the Baltic Sea is 48 mg(N) m$^{-2}$. The ship contribution to total nitrogen deposition in summer is massively reduced (by more than 60 %) in the coastal areas of the Baltic Sea region compared to 2012 (Fig. 13b). Over sea, largest reductions of the ship contribution take place in an area extending from Kattegat to the Arkona basin.

Introduction of the NECA causes a maximum reduction of the summer-accumulated nitrogen over seawater by 18 %, compared to not introducing the NECA in 2021 (Fig. 13c). This means that the Tier II fleet in "NoNECA 2040" already accomplishes a large reduction in nitrogen deposition compared to today. The effect of the lower fuel efficiency in 2040 (according to "EEDI 2040") is an increase of nitrogen deposition compared to BAU, mainly over the Northern Baltic Proper and over coastal areas. The relative increase is up to 12 % (Fig. 13d).

Table 11 shows the "BAU 2040" annual and seasonal nitrogen deposition sums to the entire Baltic Sea seawater surface, for total, oxidised and reduced nitrogen. The ship-related annual nitrogen deposition reduces by 17.6 kt N, while the overall nitrogen deposition reduces by 70.3 kt N, compared to 2012. Thus the reduction of NO$_X$ emissions over the continent, in accordance with a current legislation scenario for land-based emissions in the Baltic Sea region, has a larger impact on the future nitrogen input to the Baltic Sea than the shipping fleet.



## 6 Summary and discussion

### 6.1 Changes of the air quality in the future scenarios

In the BAU scenario, with the introduction of the NECA in 2021, $NO_X$ emissions from ship traffic in the Baltic Sea are reduced by about 80 % in 2040 because most ships of the Baltic Sea ship fleet will then fulfil the Tier III regulation. With the NoNECA

scenario, the entire ship fleet follows Tier II regulations for $NO_X$ in 2040 and, in conjunction with the fuel efficiency increase, leads to an overall $NO_X$ emission reduction from the ship fleet by about 50 %.

Table 12 presents the relative changes of annual mean concentrations of air pollutants in the Baltic Sea region between 2012 and 2040 (as average of the CD04 grid domains). Annual mean $NO_2$ decreases by 61–72 % between 2012 and 2040 in the Baltic Sea region, depending on the shipping scenario, with the smallest decrease in the NoNECA scenario.

The BAU scenario adopts the agreed $SO_X$ emission abatement regulations: the already established SECA limit of 0.1 % S in fuel from 2015 onwards followed by the global limit of 0.5 % S in ship fuels from 2020 onwards. On average, annual mean $SO_2$ decreases by ~60 % between 2012 and 2040, independent of the shipping scenario. Consequently, particulate sulphate decreases by 50–60 % over the Baltic Sea between 2012 and 2040 (not shown) in all three scenarios. The burden of $PM_{2.5}$ over the Baltic Sea region decreases by 35–37 % between 2012 and 2040 (Table 12). The reduction of $PM_{2.5}$ is larger over sea,

where it drops by 50–60 % along the main shipping routes, and smaller over the coastal areas. The large drop over sea is due to the reduction of particulate matter emissions from ships and the lower production of sulphate and nitrate related to reduced emission of primary precursor gases ($NO_X$ and $SO_X$) from ship traffic. In most coastal areas the decreased $PM_{2.5}$ is mainly a consequence of the abatement measures on land.

On annual average, the daily maximum $O_3$ decreases only slightly over the Baltic Sea region, but the summer average

decreases by 10–25 %, depending on the shipping scenario, in large parts of Sweden and the Baltic Sea, compared to present day.

Overall, a lower fuel efficiency increase than in BAU has only marginal implications on the future air quality in the Baltic Sea region.

### 6.2 Changes of the ship contribution in the future scenarios

The absolute ship contribution to ambient levels of $NO_2$ and $SO_2$ between 2012 and 2040 changes slightly more than expected due to the reduction of ship emissions. The lower abundance of $NO_X$ in the future atmospheric background increases the oxidation capacity of the atmosphere and leads to a more efficient oxidation of pollutants via gas-phase reactions and in-cloud processing. Table 13 presents the relative changes of the annual mean absolute ship contributions in the Baltic Sea region between 2012 and 2040.

A consequence of establishing the NECA is the reduction of the ship contribution to daily maximum ozone by 18 % on average compared to the present situation. If the NECA is not implemented, an increase of the ship-related daily maximum ozone by 31 % results compared to present-day. The introduction of NECA is hence critical for abating ship emissions of $NO_X$ to levels that are low enough to sustainably dampen ozone production in the Baltic Sea region. A second important effect



of the NECA over the Baltic Sea region is a reduction of secondary formation of particulate nitrate. The introduction of the NECA reduces the ship-related $PM_{2.5}$ by 72 % in 2040 compared to present-day, while it is reduced by only 48 % without implementation of the NECA.

The effect of the lower fuel efficiency on the absolute ship contribution of air pollutants is limited. Still, the annual mean
ship contributions in 2040 to the four pollutants is significantly higher than in the BAU scenario.

### 6.3   Contribution of ship emissions to nitrogen deposition

A previous study (Bartnicki et al., 2011) estimated the contribution of airborne nitrogen from international ship traffic to the oxidised nitrogen deposition in the Baltic Sea basin to be about 8 to 11 % (period: 1997–2006) on annual average. The contribution from ships with a range from 12 to 14 % has been reported for the period 2008 to 2011 (Hongisto, 2014). In the
present study the relative ship contribution to the deposition of oxidised nitrogen is 24 % (Table 6), about twice as high as the previous estimates. However, the total annual nitrogen deposition for 2012 in the present study is 29 % lower compared to the EMEP-MSC/W model used by HELCOM (Bartnicki et al., 2017). Taking the literature value of 14 % and the oxidised nitrogen deposition flux in 2012 reported by HELCOM (128.9 kt N y$^{-1}$; Bartnicki et al., 2017), an absolute ship contribution of 18 kt N y$^{-1}$ is derived, only slightly lower than our estimate of 22.5 kt N y$^{-1}$.
The relative ship contribution to the total nitrogen deposition is 14 % on annual average and 21 % in summer in the present-day situation (Table 6). The ship contribution drops to 5.6 % in 2040 (9 % in summer) when following the BAU scenario (Table 11). Between 2040 and 2012 the ship-related deposition of oxidised nitrogen decreased by 78 %. In "BAU 2040" the ship contribution to the annual deposition of oxidised nitrogen over the Baltic Sea is only 14 %.

Nitrogen deposition to the seawater of the Baltic Sea decreases on average by 40–44 % between 2012 and 2040 (Table 12).
Depending on the future shipping scenario, the decline of the ship-related nitrogen deposition varies between 46 % and 78 % (Table 13). In the EEDI scenario, when the NECA is established but fuel efficiency increase is lower than in BAU, nitrogen deposition in most ship-influenced areas decreases less than in the BAU scenario. The weakest reduction is found for the NoNECA scenario, in which nitrogen deposition decreases by only 30 % over coastal areas of Denmark, Germany and west Finland. The western part of the Baltic Sea would be most affected if the NECA is not implemented (Fig. 13c).

### 6.4   Prognosis of the total nitrogen deposition to the Baltic Sea

A linear relationship was found between the emissions of $NO_X$ from the Baltic Sea ship fleet and the annual ship-related nitrogen deposition to Baltic Sea seawater (spatial average) based on the results of the present-day simulation and the future scenario simulations (Fig. 14). Because the changes of the nitrogen deposition attributed to shipping (Fig. 13b) between 2012 and 2040 are mainly confined to the Baltic Sea and the surrounding coastal areas, it was expected that the changes of the
ship-related deposition flux are proportional to the atmospheric input of oxidised nitrogen via ship emissions. An important link between the ship emissions and the deposition of nitrogen is the formation of $HNO_3$, which constitutes the most important removal pathway for nitrogen in the atmosphere (Riemer et al., 2003).





The relationship presented above is useful for a quick evaluation of the ship-related nitrogen deposition in future shipping scenarios. Cumulative scenarios based on Shared Socioeconomic Pathways (SSPs) with respect to future ship emission in the Baltic Sea region were designed in the SHEBA project. In scenario SSP3 (regional rivalry), which represents a world with much less international trade and low mitigation capacity (Fujimori et al., 2017), future shipping deviates largely from the already decided regulations but growth of shipping is slower than in BAU by 0.5 % p.a.. The fuel efficiency development is lower by 1 % p.a. than in EEDI. Use of LNG is similar as in BAU. The Tier II regulation is not enforced in SSP3, i.e. the entire ship fleet applies Tier I standard for $NO_X$ emissions. Ship $NO_X$ emissions in SSP3 are 143 kt N y$^{-1}$, somewhat lower than in the current situation. Based on the linear model the ship-related nitrogen deposition is estimated to be 21.5 kt N y$^{-1}$.

Thus, in this quick assessment, SSP3 brings a slight improvement in 2040 compared to the current situation. The comparison of the simulated future scenarios to SSP3 also underlines the potential of the Tier II standard regulation for new built ships (as in "NoNECA 2040") to reduce the future impact from shipping; compensating, together with the faster fuel efficiency development, the projected higher ship traffic growth.

## 6.5  Discussion of uncertainties and limitations

The ship contribution to air pollutants and nitrogen deposition in the present study was computed using a zero-out method, i.e. the ship emissions were removed in one simulation. An alternative brute force method would be the perturbation of the emissions, for example reduction by 20 %, which might be more careful with respect to the non-linearity of the involved photochemistry. However, our goal was to derive the impact of shipping in different scenarios; while perturbing emissions is mainly used to investigate short-term responses to expected (small) changes of a sectoral emissions. A previous study by Geels et al. (2012) applied the so-called tagging method to assess the ship contribution from each riparian state of the Baltic Sea. Tagging requires adding auxiliary variables to the CTM itself to track pollution. While tagging for inert primary pollutants is straightforward; methods for addressing secondary pollutants requires an analysis of the limiting reagents to avoid tagging all possible follow-up products in the gas-phase, aerosol phase and cloud water. Differences between tagging and brute force methods are usually observed in secondary processes involving precursors from different sources. Some comparison studies (Collet et al., 2014; Koo et al., 2009) indicate that tagging is advantageous for source allocation rather than for predicting responses to emissions changes.

European regions that are affected by high density of ship traffic, such as UK, France, western Germany, North Sea, the southern part of the Baltic Sea and along the ship tracks in the Mediterranean are currently in a NMVOC-limited regime with respect to ozone formation (Beekmann and Vautard, 2010). In northern Europe, except of the region of the English Channel and parts of the North Sea, a transition from NMVOC-limited to $NO_X$-limited regime is projected until 2020 (Beekmann and Vautard, 2010) and the next decades (Lacressonnière et al., 2014). In a NMVOC-limited regime the production of ozone is sensitive to emissions of NMVOC, while increasing $NO_X$ leads to a reduction of ozone by titration. In the $NO_X$-limited regime, ozone is sensitive to emissions of $NO_X$ while it is hardly affected by additional NMVOC emissions.

In the simulations for the future scenarios in 2040, most certainly a transition towards a $NO_X$-limited regime happens in the currently NMVOC-limited areas of the Baltic Sea, in particular along the ship tracks in the southern part. This is clearly seen





in the "BAU 2040" scenario, where a relative increase of the ship-related daily maximum ozone occurred (due to less titration) over the central shipping routes, whereas the ship-related ozone decreased in the already $NO_X$-limited areas outside the ship tracks and over the coastal regions. However, predicted changes of the daily maximum ozone concentrations due to shipping are uncertain because of the lack of data on NMVOC emissions from shipping in the STEAM inventory that was used in the
CTM calculations.

   We have reduced land based emissions in the future scenarios in order to obtain a more realistic estimation of the consequences of regulations on shipping emissions on the future air quality in the Baltic Sea region. Based on the model results for the future ship contribution, it is obvious, that reduced land-side emissions of primary gaseous precursors amplified the decline of secondary aerosols related to shipping, in particular over the coastal areas. However, the reduction of land-side emissions
has a very small effect on the determined ship contributions to $NO_2$ and $SO_2$ over the Baltic Sea (Fig. S13).

   The reason for the underestimation of $WNO_3$ and $WNH_4$ in the CMAQ simulations, compared to observations of the regional background monitoring stations of the EMEP network, could not be fully resolved. The formation of particulate nitrate involves complex chemistry of several compounds in the gas-phase and multicomponent solution systems on aerosols. The simulation of nitrate is highly uncertain because it requires accurate computation of the concentrations of the precursors, e.g. $HNO_3$, $NH_3$,
dust and sea-salt. The joint underestimation of $WNO_3$ and $WNH_4$ was found in the statistical analysis of model-observation pairs and also in the comparison of modelled and observed seasonal averages. The most convincing explanation at the current stage is, that the oxidative conversion of $NO_X$ to $HNO_3$ occurs at a too slow rate in the model, combined with too little particulate ammonium from the regional background that is advected into the Baltic Sea region.

   An alternative explanation might be that the wet removal of $NO_3^-$ and $NH_4^+$ in CMAQ is not efficient enough. In addition,
the evaluation of simulated precipitation amounts and frequency showed that the southern part of the Baltic Sea receives too little rainfall in summer. For the other seasons and in the northern part the precipitation bias is positive. Too low precipitation in the southern part, where modelled concentrations of $NO_3^-$ and $NH_4^+$ are much higher compared to the northern part, could be responsible for an average underestimation of the total nitrogen wet deposition to the Baltic Sea.

   Coarse mode particles are much faster removed than fine mode particles, therefore the deposition of particulate nitrate
crucially depends on the uptake to larger particles. Heterogeneous chemical production of nitrate on coarse mode particles has been found to control the atmospheric nitrate production to a very large extent (Bian et al., 2017). The hydrolysis of $N_2O_5$ to produce $HNO_3$ is considered in CMAQ by uptake coefficients depending on temperature, RH and particle composition, using the parameterization by Davis et al. (2008), but only for fine-mode aerosols. The Davis parameterization tends to predict too high $N_2O_5$ uptake coefficients near the surface, especially over marine and coastal areas where relative humidity is high (Chang
et al., 2016). CMAQ allows for a dynamic mass transfer to coarse particles and therefore takes into account the reactive uptake of $HNO_3$ by sea salt particles. Meanwhile, resuspension of mineral dust was not activated in the simulations, and the missing heterogeneous reaction on dust particles surfaces may have contributed to the underestimation of $WNO_3$.





## 7 Conclusions

The impact of ship emissions on the present-day (2012) and future (2040) air quality and nitrogen deposition was evaluated with a regional atmospheric CTM. The meteorological fields, the emissions from ship traffic and the emissions from land-based sources are considered at a grid resolution of $4 \times 4\,\mathrm{km}^2$ for the inner-most model domain covering most of the Baltic Sea

region. Ship emissions from the STEAM model based on ship movements from AIS records and detailed ship characteristics in combination with solving atmospheric chemistry and transport at high resolution, enable a better treatment of the plumes from ship traffic, compared to previous CTM studies in the Baltic Sea region.

The effect of future legislation related to shipping and of future changes of the ship fuel efficiency of the ship fleet on air quality and deposition in 2040 in the Baltic Sea region was determined based on computational results from regional CTM

simulations. Future air quality and nitrogen deposition is analysed, in order to investigate: (1) the effect of establishing the NECA in 2021 compared to a future situation without NECA; and (2) the effect of a lower fuel efficiency increase than expected based on continuation of the current trend. A BAU scenario has been designed in which the NECA is implemented and the fuel efficiency for new ships improves more than required by IMO's Energy Efficiency Design Index regulation.

Establishing the NECA in 2021 has several benefits for the Baltic Sea environment. One important effect of the NECA is a

15 reduction of secondary formation of particulate nitrate. The introduction of the NECA reduces the ship-related $PM_{2.5}$ by 72 % in 2040 compared to present-day, while it is reduced by only 48 % without implementation of the NECA. A major consequence of establishing the NECA is a reduction of the ship contribution to daily maximum ozone in 2040 compared to the present situation. If the NECA is not implemented, an increase of the ship-related daily maximum ozone results compared to present-day. The introduction of NECA is thus critical for abating ship emissions of $NO_X$ to levels that are low enough to sustainably

dampen ozone production in the Baltic Sea region. Overall, the introduction of the NECA is expected to be beneficial for avoiding future health impacts of ozone and $PM_{2.5}$ in coastal areas of the southern part of the Baltic Sea region.

The effect of the lower fuel efficiency on the absolute ship contribution of air pollutants is relatively small. The implementation of the NECA in 2021 can be regarded as safeguard for the case that the fuel efficiency increase falls below the projected trend.

The decline of the ship-related nitrogen deposition to the Baltic Sea between 2012 and 2040 varies between 46 % and 78 % in the different future scenarios. When the NECA is established but the fuel efficiency increase is lower than expected, nitrogen deposition in most ship-influenced areas decreases less than in the BAU scenario. The weakest reduction is found for the scenario without implementing the NECA, in which nitrogen deposition decreases by only 30 % over coastal areas of Denmark, Germany and west Finland. The western part of the Baltic Sea would be most affected if the NECA is not

implemented.

A prognostic relationship for a quick evaluation of the ship-related nitrogen deposition in future shipping scenarios was derived in this work. The relationship should be further modified to consider the inter-annual variability of atmospheric deposition due to changing meteorological conditions in order to allow for more robust projections of the ship-related nitrogen input





to the Baltic Sea. However, it may be used for estimating possible exceedances of critical loads for eutrophying substances that are based on annual nitrogen inputs.

A limitation of the model results for regional surface concentrations of the daily maximum ozone concentrations over the Baltic Sea region is the lack of data on NMVOC emissions from shipping in the STEAM inventory that was used in the CTM

calculations. Additional NMVOC emissions from shipping would serve as precursors of ozone and enhance photochemical ozone production in a NMVOC-limited regime. In the presented model simulations, $NO_X$ emissions from continental sources were reduced by 60 % between 2012 and 2040, following current legislation, i.e. already decided emission abatement regulations. The lower abundance of $NO_X$ in the future could lead to a shift in the overall atmospheric chemical regime. To predict more accurately how such change in the chemical regime will affect the future influence of ship emissions, a better handle on

NMVOC emissions from ships and their future development would be important.

As a consequence of $SO_X$ emission abatement regulations for shipping, annual mean $SO_2$ decreases on average by ~60 % between 2012 and 2040, independent of the future scenario. With the reduction of $SO_2$ emissions, less $NH_3$ is required to neutralise the strong acid $H_2SO_4$. The excess $NH_3$ is available for the formation of $NO_3^-$ and $NH_4^+$ in the particulate phase. According to Tsimpidi et al. (2008), the trend of future particulate $NO_3^-$ concentrations depends on whether $NO_X$ or $NH_3$ are

15 the limiting gas-phase compounds for nitrate formation. Measurements in southern Sweden have shown that the concentrations of $NH_3$ and $HNO_3$ are too low to form pure solid or aqueous ammonium nitrate particles (Ferm, 1992). Thus in a future background atmosphere over the Baltic Sea region, ambient levels of both gases might be too low for ammonium nitrate formation, and the fate of these gases would be the removal by dry and wet deposition. Meanwhile, the formulation of heterogeneous processes related to the production of nitrate are highly uncertain in the models, limiting the conclusions about the future

transition in the nitrate formation regime.

A related study by Jutterström et al. (2018) assessed the extent of environmental damage related to shipping on the terrestrial ecosystems surrounding the Baltic Sea. Ecological impacts of air pollutants on land are evaluated in terms of critical load (CL) exceedance for eutrophication. Using the latest reported CL values for eutrophication together with the modelled deposition data of nitrogen for 2012 and the future scenarios for 2040 of the present study, Jutterström et al. (2018) find a significant

improvement from 2012 to 2040. For the BAU scenario, the area where the CL (eutrophication) are exceeded due to ship-related nitrogen deposition decreased from about 20 % in 2012 to 5 % in 2040. If the NECA is not implemented, the exceeded area due to shipping is about 14 % in 2040, indicative for the relevance of the NECA for coastal ecosystems surrounding the Baltic Sea. We note, that the use of gridded model data of dry deposition in the estimation of CL exceedances has limitations. In the model simulation, dry deposition to land surfaces is weighted for the different land use classes present in each grid cell.

This might lead to an underestimation of the eutrophication risk for forests in a grid cell which includes other land uses, as the canopy resistance of forests is much higher than that of grassland or other low vegetation. The CMAQ deposition data is less affected by this problem due the high resolution of the gridded data.

The shipping sector is an important contributor to atmospheric nitrogen deposition to the Baltic Sea. The present study estimates a deposition flux of oxidised nitrogen in the order of 22.5 kt N $y^{-1}$ due to shipping emissions for the year 2012,

slightly higher than previous estimates (Hongisto, 2014: Bartnicki et al., 2017). Occurrences of high nutrient input to coastal



waters have been suggested to cause short-term algal blooms (Spokes et al., 2000). On the other hand, a study in the Kattegat showed that direct nitrogen inputs through atmospheric deposition could not be linked to any summer algal bloom observation, probably because the atmospheric input is considerably diluted through mixing in the surface water layer (Carstensen et al., 2005). The incidence of harmful algal blooms, which cause health damages to humans and animals in shallow coastal waters,

has also been linked to atmospheric nitrogen inputs (Paerl, 1997). However, the relationships between high nutrient inputs and the development of harmful algal blooms are still not well understood (Anderson et al., 2002).

Much stricter regulations for $NO_X$ emission from new built ships will be enforced in 2021. It can be expected that significant emission reductions will be the consequence of these regulations, however, this requires that the exhaust gas cleaning technologies that will be implemented on board of most the new built ships work properly. From the experiences with Euro 4 and Euro 5

diesel cars that frequently emit much more $NO_X$ than allowed, policy should pave the way for extended compliance control measures. Several techniques exist how emissions from ships can be measured, including in-situ observations at coastlines, ground based remote sensing techniques, sniffers on board of aircraft or drones and sensors on board of the ships. The best technology needs to be tested now in order to be prepared for the implementation of the NECA.

*Data availability.* The COSMO-CLM precipitation data and the CMAQ data on air pollutant concentrations and nitrogen deposition are

available upon request.

*Author contributions.*

**Matthias Karl:** overall structure; input data preparation; CMAQ simulation; evaluation of the air concentrations and nitrogen deposition data; programming; framework for data processing; visualisation and plotting; major writing tasks

**Johannes Bieser:** land-based emissions with SMOKE-EU; text contribution on present-day and future land-based emissions

**Beate Geyer:** COSMO-CLM model simulations; evaluation of precipitation data; visualisation of precipitation data; text contributions on meteorological modelling

**Volker Matthias:** development of research questions; contribution to the Conclusions; discussions of the manuscript structure

**Jukka-Pekka Jalkanen:** design of future scenarios; text contribution on shipping emissions

**Lasse Johansson:** shipping emissions with STEAM; text contribution on shipping emissions

**Erik Fridell:** development of the future scenarios; data and text contribution on scenarios

*Competing interests.* The authors declare that they have no conflict of interest

*Acknowledgements.* This work is part of the BONUS SHEBA (Sustainable Shipping and Environment of the Baltic Sea region) research project under Call 2014-41. BONUS (Art 185) is funded jointly by the EU, Innovation Fund Denmark, Estonian Research Council, Academy



of Finland, Forschungszentrum Jülich Beteiligungsgesellschaft mbH (Germany), National Centre of Research and Development (Poland) and Swedish Environmental Protection Agency.

The air quality model CMAQ is developed and maintained by the U.S. Environmental Protection Agency (US EPA). COSMO-CLM is the community model of the German climate research (www.clm-community.eu). The simulations with COSMO-CLM and CMAQ were

5    performed at the German Climate Computing Centre (DKRZ) within the project "Regional Atmospheric Modelling" (Project Id 0302). The Swedish Meteorological and Hydrological Institute (SMHI) is thanked for making available the precipitation data from rain gauge stations in Sweden.

Z. Klimont (IIASA) is thanked for emission data for the 2040 CLE scenario from ECLIPSE v5. NILU is thanked for the EBAS database maintenance and data provision. Sara Jutterström (IVL) is thanked for good collaboration and discussion of model results on deposition of

10   nitrogen and sulphur.



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



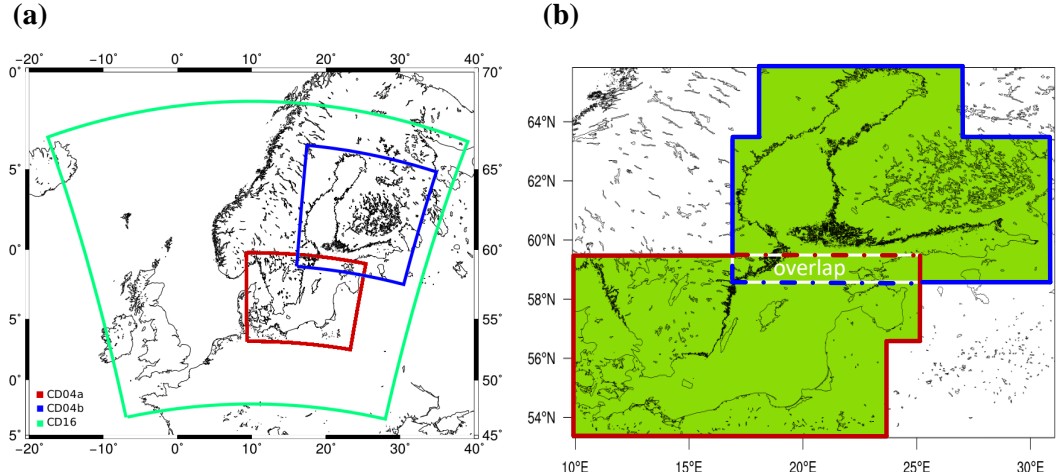

**Figure 1.** Model nests used in the simulations with CMAQ and for the spatial maps of model results: (a) computational grid for Northern Europe with $16 \times 16 \, \text{km}^2$ resolution (CD16, green) and the high-resolution grids of $4 \times 4 \, \text{km}^2$ for southern Baltic Sea (CD04a, dark red) and northern Baltic Sea (CD04b, dark blue). (b) Exemplary structure of spatial maps spanning from latitude $53.30°$ N (south) to $65.80°$ N (north) and longitude $9.85°$ E (west) to $30.95°$ E (east). Green shaded area is the high-resolution area which shows output from regional model runs with a grid resolution of $4 \times 4 \, \text{km}^2$. Dark red outline marks the extent of the southern part of the Baltic Sea region and dark blue outline marks the extent of the northern part of the Baltic Sea region, for which model output from two high-resolution nests were used. For the overlap area, the arithmetic mean of results from both nests was used. In the post-processing of model results, the native Lambert conformal projection of CMAQ output was transformed to a regular lat-lon grid, therefore the two outlined areas do not fill complete rectangles. The entire domain shown in (b) was interpolated to a uniform resolution of $0.05°$ in the post-processing. White areas of the map are covered by the output from the model nest with $16 \times 16 \, \text{km}^2$ resolution.



**(a)**                                                    **(b)**

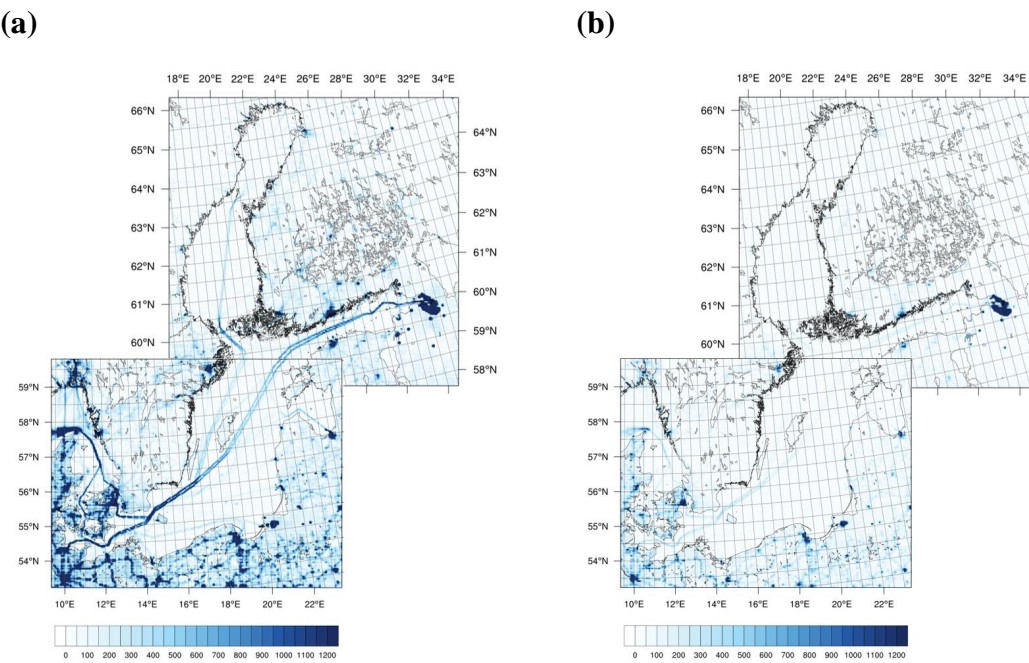

**Figure 2.** Annual total emissions of $NO_X$ $(mg(N) \, m^{-2})$ in the Baltic Sea region: (a) in 2012 and (b) in 2040 for the BAU scenario. Gridded emissions from the STEAM and SMOKE emission databases interpolated to a grid resolution of $4 \times 4 \, km^2$ and transformed to Lambert conformal projection for the two CMAQ high-resolution domains. Grid lines mark a lat-lon grid with $0.5 \times 0.5$ degrees cells.





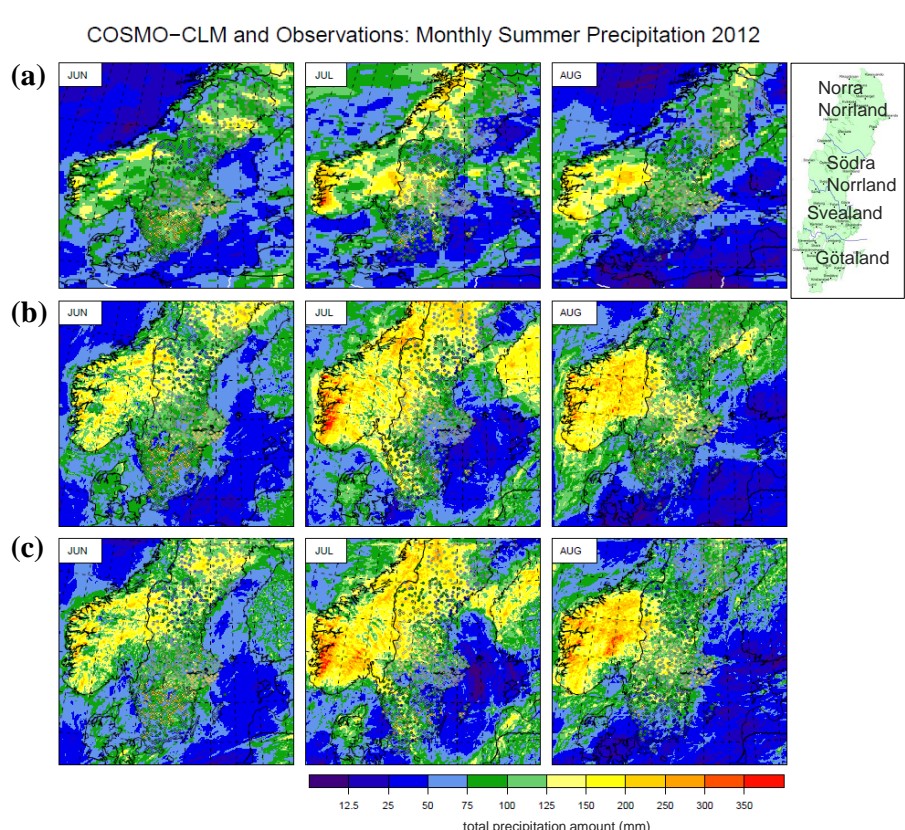

**Figure 3.** Monthly accumulated precipitation (mm) maps for summer months (JJA) in 2012: computed precipitation fields from different configurations of COSMO-CLM compared to observations at meteorological stations shown as circles filled with corresponding colour of observed precipitation: (a) COSMO-CLM on 0.11 degrees ("011"), (b) COSMO-CLM on 0.025 degrees with parameterised convection ("0025_Tiedtke") and (c) COSMO-CLM on 0.025 degrees with convection-permitting configuration ("0025_convper"), as used for the CMAQ high resolution domain. Observation data is based on precipitation measurements at 1804 stations of the Swedish meteorological network from SMHI. Inset in the top right corner shows the definition of the four regions of Sweden with blue border lines.

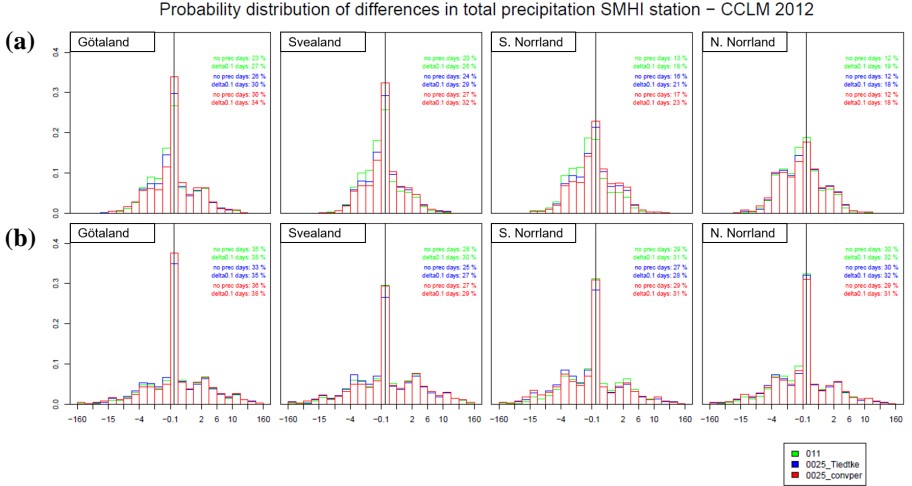

**Figure 4.** Probability distribution of the differences in daily precipitation sums (mm d$^{-1}$) between the SMHI station observations and COSMO-CLM with different configurations ("011", "0025_Tiedtke", and "0025_convper") in the four regions of Sweden (from left to right: Götaland, Svealand, Södra Norrland, Norra Norland): (a) for the winter months (JFD) of 2012 and (b) for the summer months (JJA) of 2012. The percentage fraction of days with zero difference between model and observation ("no prec." days) and the percentage fraction of days with difference of ±0.1 mm d$^{-1}$ ("delta0.1" days) is indicated in the plots for each model configuration.





**(a)**

**(d)** Leba NO3

**(e)** Virolahti II NO3

**(b)** Zingst NO3

**(f)** Preila NO3

**(c)** Råö NO3

**(g)** Ähtäri NO3

**Figure 5.** Comparison of modelled wet deposition of nitrate (WNO$_3$) as daily sums (in mg(N) m$^{-2}$ d$^{-1}$) from the 16-km resolution grid (red) and 4km-resolution grid (blue) to observed daily sums (black crosses) at regional background stations around the Baltic Sea from the EMEP monitoring network: (a) map with stations as red circles, (b) Zingst, DE0009R, (c) Råö, SE0014R, (d) Leba, PL0004R, (e) Virolahti II, FI0017R, (f) Preila, LT0015R, and (g) Ähtäri, FI0004R. Comparison time period: 1 March to 30 November 2012. All available data from simulations and observations are shown.



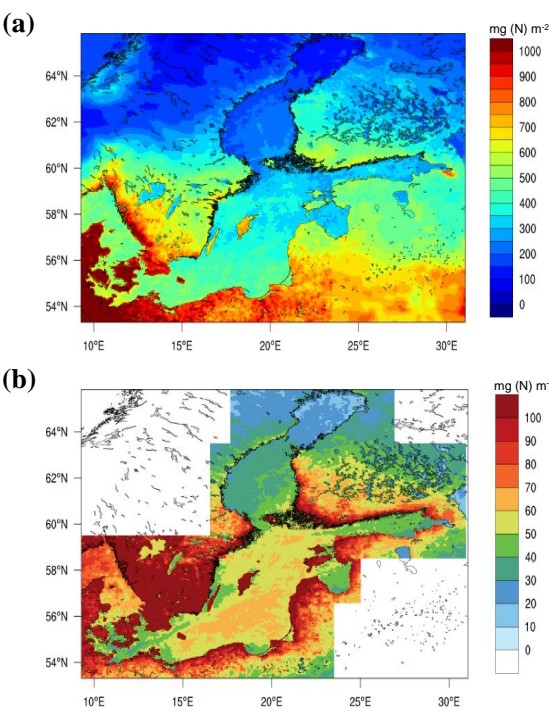

**Figure 6.** Present-day (2012) accumulated total deposition of nitrogen (in mg(N) m$^{-2}$) in the Baltic Sea region from CMAQ model results: (a) annual deposition and (b) annual ship-related deposition. Ship contribution is only shown for the high-resolution area.







**Figure 7.** Present-day (2012) ship contribution in the Baltic Sea region in summer (JJA) from CMAQ model results: ship-related concentration (left) for gaseous pollutants (in ppbv) and for $PM_{2.5}$ (in $\mu g\,m^{-3}$), percentage ship contribution (right) for (a) daily maximum $O_3$, (b) $NO_2$, (c) $SO_2$, and (d) $PM_{2.5}$. Ship-related contribution only shown for the high-resolution area. See text for details.



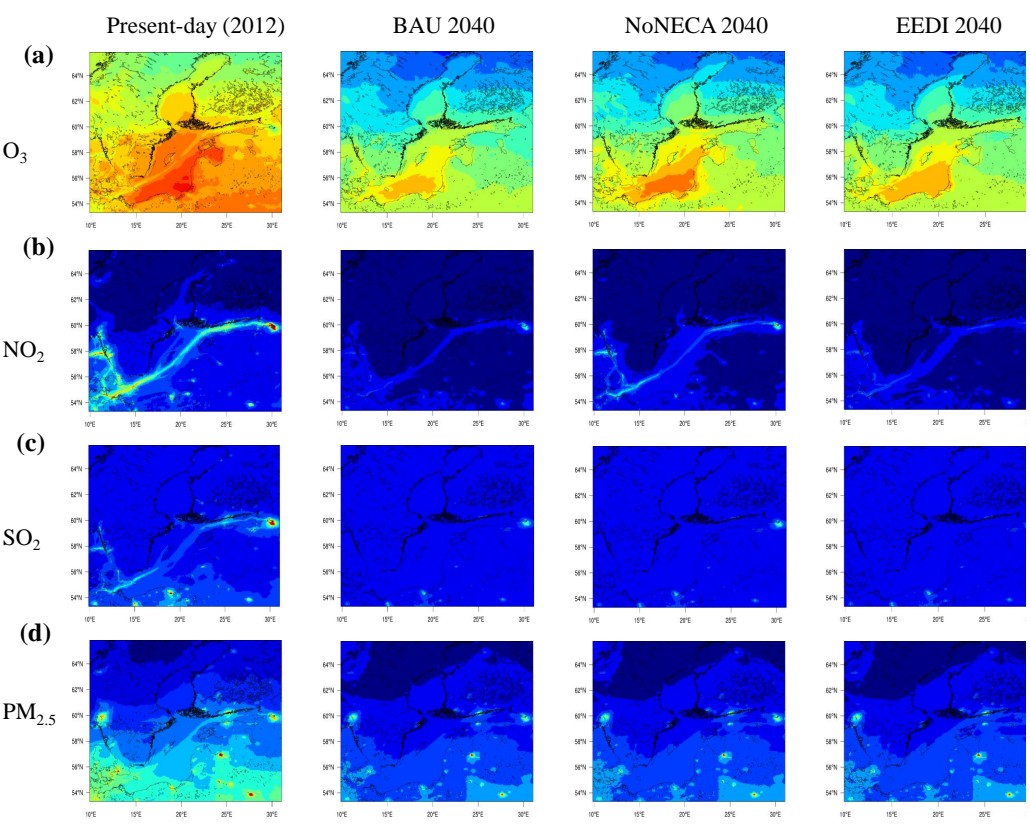

**Figure 8.** Future air quality situation in the Baltic Sea region in summer (JJA) compared to present-day. CMAQ model results for present-day (first column), for "BAU 2040" (second column), for "NoNECA 2040" (third column), and for "EEDI 2040" (fourth column), are shown for (a) daily maximum $O_3$, (b) $NO_2$, (c) $SO_2$, and (d) $PM_{2.5}$.







**Figure 9.** Future (2040) ship contribution in the Baltic Sea region in summer (JJA) from CMAQ model results for the "BAU 2040" scenario: ship-related concentration (left) for gaseous pollutants (in ppbv) and for $PM_{2.5}$ (in $\mu g\,m^{-3}$), percentage ship contribution (right) for (a) daily maximum $O_3$, (b) $NO_2$, (c) $SO_2$, and (d) $PM_{2.5}$. Ship-related contribution only shown for the high-resolution area. Same scales as in Figure 7 were used to facilitate comparison of the concentration and contribution maps. The sharp change of the $O_3$ ship contribution north of 58.8° N is an artefact of the averaging in the overlap area of the two 4-km resolution grids.





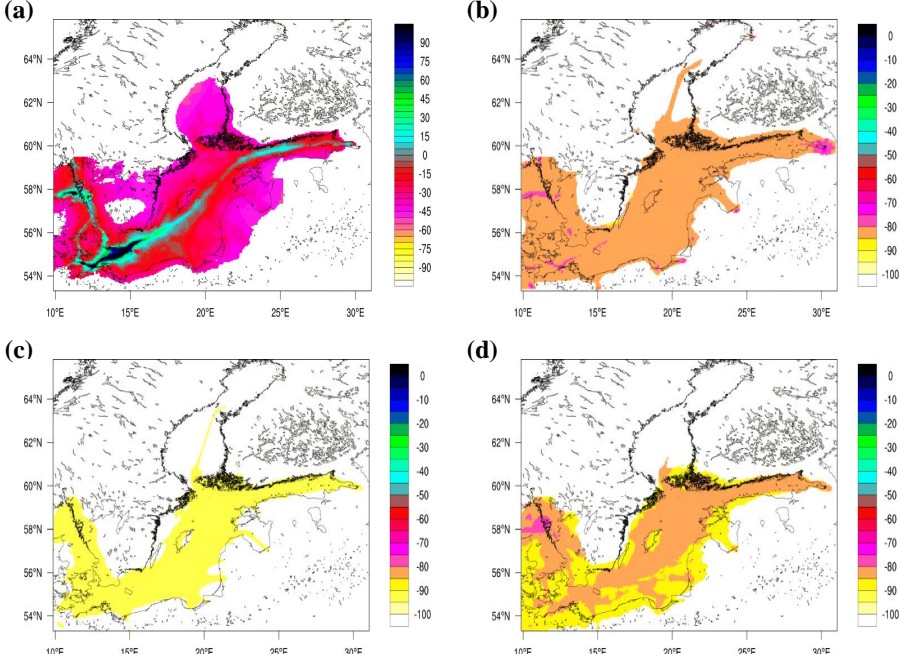

**Figure 10.** Future (2040) change of the ship-related contribution in summer (JJA) in percent compared to 2012, given as relative difference between the ship contribution from the "BAU 2040" simulation and the ship contribution from the present-day simulation: (a) daily maximum $O_3$, (b) $NO_2$, (c) $SO_2$, (d) $PM_{2.5}$. Not coloured (empty) areas indicate grid cells with ship contribution in "BAU 2040" of less than 1.0 ppbv, 0.1 ppbv, 0.01 ppbv, 0.005 µg m$^{-3}$, for daily max. $O_3$, $NO_2$, $SO_2$, $PM_{2.5}$, respectively. Ship-related contribution only shown for the high-resolution area. Note the different scale for daily max. $O_3$ (from -100 % to 100 %).

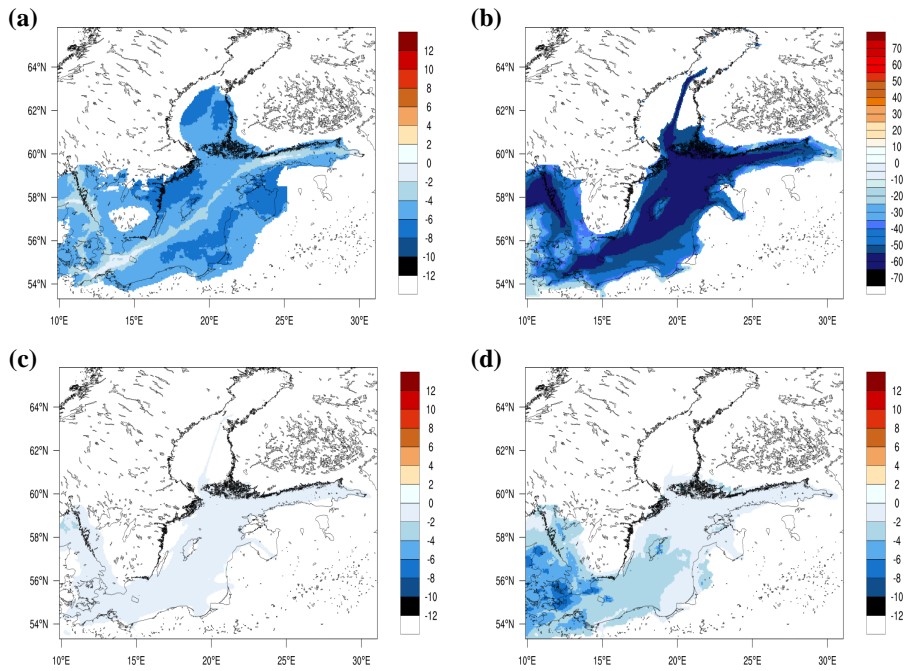

**Figure 11.** Effect of establishing the NECA (in 2021) on the future air quality in summer (JJA) 2040 in the Baltic Sea region as relative difference (in percent) between the scenario simulations "BAU 2040" and "NoNECA 2040": (a) daily maximum $O_3$, (b) $NO_2$, (c) $SO_2$, (d) $PM_{2.5}$. Not coloured (white) areas indicate grid cells with ship contribution in "BAU 2040" of less than 1.0 ppbv, 0.1 ppbv, 0.01 ppbv, 0.005 µg m$^{-3}$, for daily max. $O_3$, $NO_2$, $SO_2$, $PM_{2.5}$, respectively.



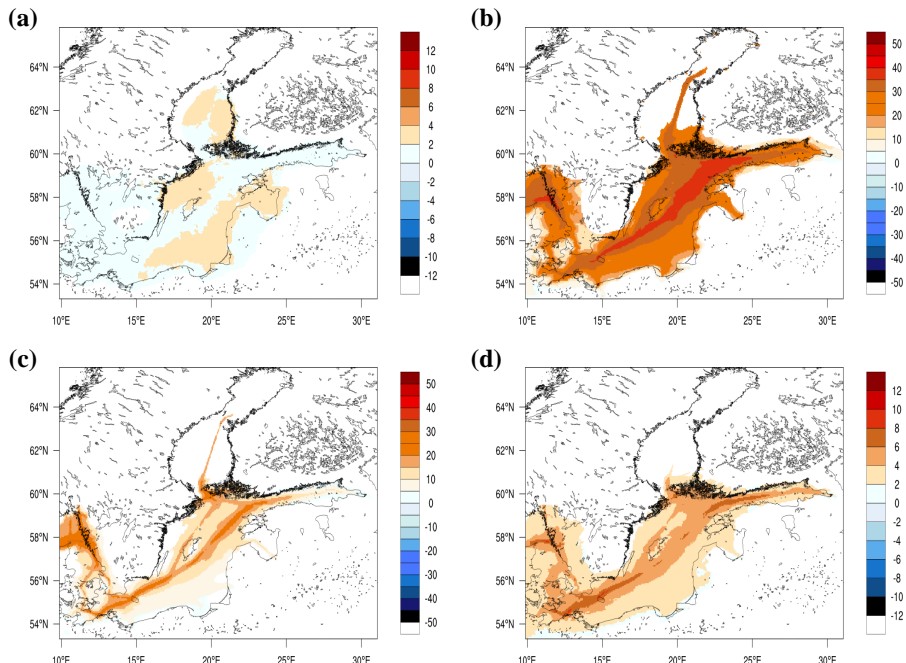

**Figure 12.** Effect of lower fuel efficiency on the future air quality in summer (JJA) 2040 in the Baltic Sea region as relative difference (in percent) between the scenario simulations "EEDI 2040" and "BAU 2040": (a) daily maximum $O_3$, (b) $NO_2$, (c) $SO_2$, (d) $PM_{2.5}$. Not coloured (white) areas indicate grid cells with ship contribution in "BAU 2040" of less than $1.0$ ppbv, $0.1$ ppbv, $0.01$ ppbv, $0.005\,\mu g\,m^{-3}$, for daily max. $O_3$, $NO_2$, $SO_2$, $PM_{2.5}$, respectively.





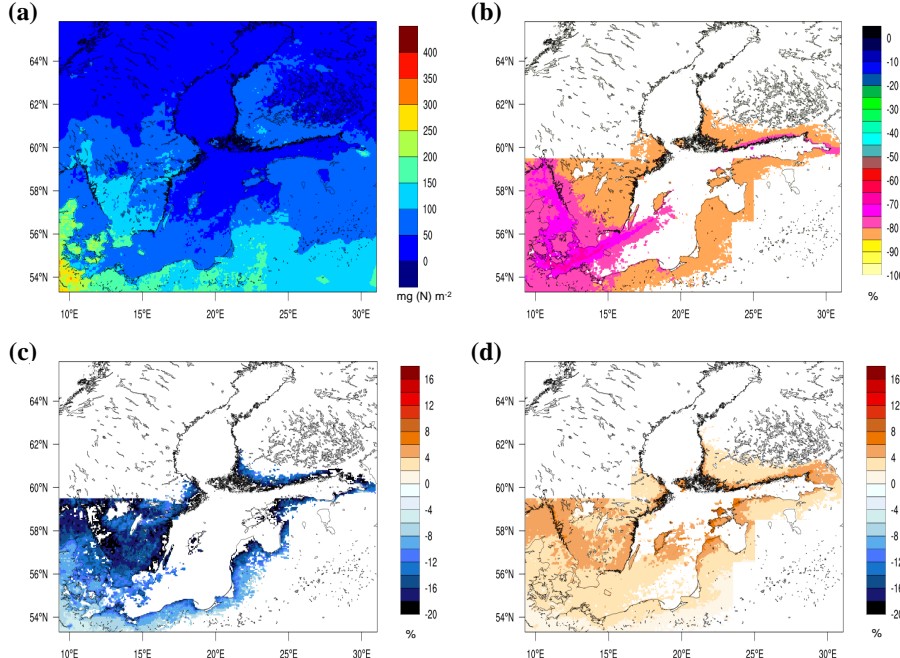

**Figure 13.** Nitrogen deposition in summer (JJA) 2040: a) accumulated total deposition of nitrogen (in mg(N) m$^{-2}$) in scenario "BAU 2040", b) percentage change of the ship contribution to nitrogen deposition in scenario "BAU 2040" compared to present day, c) effect of the NECA on nitrogen deposition, and d) effect of the lower efficiency of EEDI on nitrogen deposition. Not coloured (empty) areas indicate grid cells with ship contribution in "BAU 2040" of less than 6.0 mg(N) m$^{-2}$ for total nitrogen deposition. Ship-related contribution only shown for the high-resolution area.





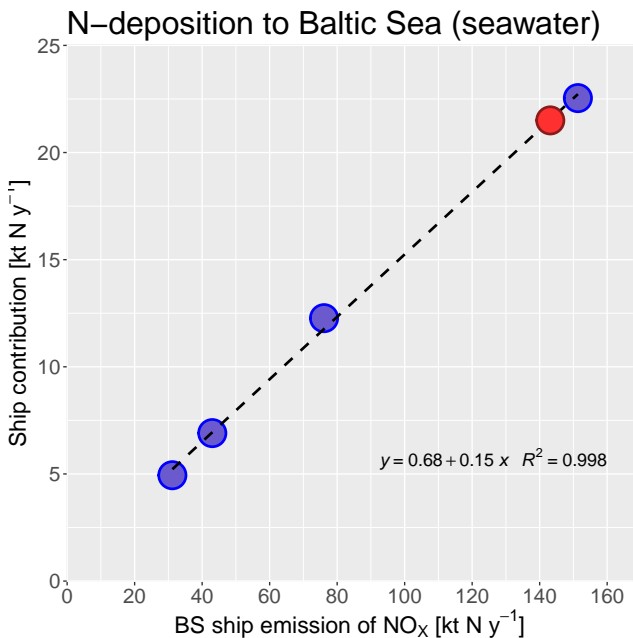

**Figure 14.** Relationship between emissions of $NO_X$ (in $kt\,N\,y^{-1}$) from the Baltic Sea ship fleet and the annual ship-related nitrogen deposition (in $kt\,N\,y^{-1}$) to the Baltic seawater (on spatial average) based on the model results of the present-day simulation and the model results of the future scenario simulations. Red filled circle indicates the ship contribution in scenario SSP3 predicted from the linear fit to the relationship.



**Table 1.** Overview of SMOKE-EU source sectors. International shipping refers to shipping outside North and Baltic seas.

| SNAP | Description | Source type | Inventory |
|------|-------------|-------------|-----------|
| 1 | Energy and heat production | point | EPER |
| 2 | Residential combustion | area | EMEP |
| 3 | Industrial combustion | point | EPER |
| 4 | Manufacturing processes | point | EPER |
| 5 | Refineries | point | EPER |
| 6 | Product use | area | EMEP |
| 7 | On road emissions | line | EMEP |
| 8.1 | Off road emissions | area | EMEP |
| 8.2 | Inland shipping | line | EMEP |
| 8.3 | Aviation | area | EMEP |
| 8.4 | International shipping | area | EMEP |
| 9 | Waste incineration | point | EPER |
| 10.1 | Agriculture | area | EDGAR |
| 10.2 | Animal husbandry | area | EDGAR |

**Table 2.** Future scenario emissions: emission scaling factors used in the three scenarios for shipping emissions for the relevant air pollutants. PM-other includes EC, OC and mineral ash. The emission scaling factors give the respective emissions in 2040 in relation to the emissions in 2012.

| Scenario | CO | PM-other | $SO_4$ | $SO_X$ | $NO_X$ |
|----------|------|----------|--------|--------|--------|
| BAU | 0.679 | 0.351 | 0.088 | 0.088 | 0.207 |
| NoNECA | 0.679 | 0.351 | 0.088 | 0.088 | 0.505 |
| EEDI | 0.923 | 0.490 | 0.121 | 0.207 | 0.285 |

**Table 3.** Precipitation bias, calculated as difference between model and observation (M-O; in mm) for summer (JJA) and for winter (JFD) of 2012. Values are given as average of all stations in a region.

| Region | Precipitation bias (M-O) [mm] in summer | | | Precipitation bias (M-O) [mm] in winter | | |
|--------|------|--------|---------|------|--------|---------|
| | 011 | Tiedtke | convper | 011 | Tiedtke | convper |
| Götaland | -99 | -35 | -67 | -1 | 0 | -8 |
| Svealand | -83 | -40 | -61 | 20 | -4 | -11 |
| S. Norrland | 10 | 111 | 132 | 50 | 18 | 5 |
| N. Norrland | 49 | 89 | 101 | 43 | 34 | 53 |



**Table 4.** Statistical comparison of daily sums of WNO$_3$ for stations of the EMEP monitoring network in the Baltic Sea region. CMAQ model results for the CD16 and CD04 grid domains are evaluated separately. Statistical indicators include mean values of model ($\mu_{\text{Mod}}$), mean values of observations ($\mu_{\text{Obs}}$), Spearman's correlation coefficient (R$_{\text{Spr}}$), and normalized mean bias (NMB). Unit of mean values is mg(N) m$^{-2}$ d$^{-1}$. $N$ is the number of samples where precipitation occurred both in the simulation and in the observation. Only stations with more than 7 samples were considered (for station Hailuoto II it is less than 7 for the CD04 grid).

| Station, Code | CD16 | | | | | CD04 | | | | |
|---|---|---|---|---|---|---|---|---|---|---|
| | $N$ | R$_{\text{Spr}}$ | $\mu_{\text{Mod}}$ | $\mu_{\text{Obs}}$ | NMB | $N$ | R$_{\text{Spr}}$ | $\mu_{\text{Mod}}$ | $\mu_{\text{Obs}}$ | NMB |
| Zingst, DE0009R | 14 | -0.09 | 0.65 | 5.23 | -0.88 | 9 | 0.08 | 1.21 | 6.03 | -0.80 |
| Råö, SE0014R | 86 | 0.49 | 2.37 | 3.21 | -0.26 | 77 | 0.55 | 1.76 | 2.95 | -0.40 |
| Leba, PL0004R | 72 | 0.26 | 1.56 | 3.41 | -0.54 | 58 | 0.40 | 1.48 | 3.78 | -0.61 |
| Diabla Gora, PL0005R | 75 | 0.29 | 1.25 | 2.42 | -0.48 | 63 | 0.01 | 1.61 | 3.02 | -0.47 |
| Ähtäri, FI0004R | 12 | 0.06 | 0.75 | 3.57 | -0.79 | 11 | 0.81 | 0.78 | 3.16 | -0.75 |
| Virolahti II, FI0017R | 17 | -0.01 | 0.92 | 5.93 | -0.85 | 14 | -0.17 | 1.07 | 5.42 | -0.80 |
| Hailuoto II, FI0053R | 13 | 0.32 | 0.69 | 3.07 | -0.78 | 5 | – | – | – | – |
| Lahemaa, EE0009R | 70 | 0.31 | 1.05 | 1.66 | -0.37 | 50 | 0.27 | 1.10 | 1.84 | -0.40 |
| Preila, LT0015R | 58 | 0.32 | 1.46 | 2.66 | -0.45 | 44 | 0.40 | 2.04 | 2.733 | -0.25 |

**Table 5.** Statistical comparison of daily sums of WNH$_4$ for stations of the EMEP monitoring network in the Baltic Sea region. CMAQ model results for the CD16 and CD04 grid domains are evaluated separately. Statistical indicators include mean values of model ($\mu_{\text{Mod}}$), mean values of observations ($\mu_{\text{Obs}}$), Spearman's correlation coefficient (R$_{\text{Spr}}$), and normalized mean bias (NMB). Unit of mean values is mg(N) m$^{-2}$ d$^{-1}$. $N$ is the number of samples where precipitation occurred both in the simulation and in the observation. Only stations with more than 7 samples were considered (for station Hailuoto II it is less than 7 for the CD04 grid).

| Station, Code | CD16 | | | | | CD04 | | | | |
|---|---|---|---|---|---|---|---|---|---|---|
| | $N$ | R$_{\text{Spr}}$ | $\mu_{\text{Mod}}$ | $\mu_{\text{Obs}}$ | NMB | $N$ | R$_{\text{Spr}}$ | $\mu_{\text{Mod}}$ | $\mu_{\text{Obs}}$ | NMB |
| Zingst, DE0009R | 12 | 0.23 | 0.76 | 8.88 | -0.91 | 8 | 0.52 | 1.54 | 9.60 | -0.84 |
| Råö, SE0014R | 81 | 0.52 | 2.03 | 3.28 | -0.38 | 63 | 0.38 | 1.83 | 3.42 | -0.46 |
| Leba, PL0004R | 69 | 0.35 | 1.31 | 4.21 | -0.69 | 57 | 0.24 | 1.29 | 4.64 | -0.72 |
| Diabla Gora, PL0005R | 68 | 0.20 | 1.28 | 3.39 | -0.62 | 56 | 0.06 | 1.86 | 3.77 | -0.51 |
| Ähtäri, FI0004R | 12 | 0.07 | 0.68 | 2.59 | -0.74 | 9 | 0.58 | 0.81 | 2.82 | -0.71 |
| Virolahti II, FI0017R | 18 | -0.07 | 0.70 | 4.95 | -0.86 | 13 | -0.03 | 1.02 | 4.68 | -0.78 |
| Hailuoto II, FI0053R | 13 | -0.18 | 0.60 | 3.50 | -0.83 | 4 | – | – | – | – |
| Lahemaa, EE0009R | 47 | 0.46 | 1.09 | 1.43 | -0.24 | 36 | 0.29 | 1.12 | 1.76 | -0.37 |
| Preila, LT0015R | 53 | 0.48 | 1.23 | 2.26 | -0.45 | 40 | 0.45 | 2.04 | 2.53 | -0.19 |





**Table 6.** Present-day annual and seasonal nitrogen deposition amounts (kt N) to the seawater of the Baltic Sea for 2012 and ship-related nitrogen deposition from the CD04 grid. Amounts refer to a Baltic Sea surface area of 431390 km$^2$, including the western part of Skagerrak.

| Nitrogen deposition | | Year | JFD | MAM | JJA | SON |
|---|---|---|---|---|---|---|
| All emissions | Oxidised | 94.5 | 23.1 | 16.1 | 23.1 | 32.1 |
| | Reduced | 64.5 | 9.1 | 18.3 | 17.5 | 19.5 |
| | Total | 159.0 | 32.2 | 34.5 | 40.6 | 51.7 |
| Ship emissions | Total | 22.5 | 3.9 | 4.3 | 8.5 | 5.8 |

**Table 7.** Statistical evaluation of modelled O$_3$ concentrations (in ppbv) with measurements of the EMEP monitoring network in the Baltic Sea region based on daily means for the entire year and for summer (JJA). Statistical indicators: mean values of model ($\mu_{\mathrm{Mod}}$), mean values of observations ($\mu_{\mathrm{Obs}}$), Spearman's correlation coefficient (R$_{\mathrm{Spr}}$), normalized mean bias (NMB) and root mean square error of the modelled values (RMSE; in ppbv). All stations of the EMEP network located within the CD04 grid domain with available measurements were considered. $N$ is the number of the available daily mean measurements at the respective station.

| Station, Code | CD04 annual | | | | | | CD04 summer | | | | | |
|---|---|---|---|---|---|---|---|---|---|---|---|---|
| | $N$ | R$_{\mathrm{Spr}}$ | $\mu_{\mathrm{Mod}}$ | $\mu_{\mathrm{Obs}}$ | NMB | RMSE | $N$ | R$_{\mathrm{Spr}}$ | $\mu_{\mathrm{Mod}}$ | $\mu_{\mathrm{Obs}}$ | NMB | RMSE |
| Zingst, DE0009R | 366 | 0.80 | 27.3 | 27.8 | -0.02 | 5.9 | 92 | 0.64 | 29.0 | 33.0 | -0.12 | 6.5 |
| Keldsnor, DK0005R | 357 | 0.66 | 26.3 | 28.1 | -0.07 | 7.2 | 86 | 0.29 | 26.8 | 32.9 | -0.18 | 9.7 |
| Risoe, DK0012R | 366 | 0.74 | 26.1 | 29.3 | -0.11 | 6.1 | 92 | 0.48 | 26.9 | 35.0 | -0.23 | 10.0 |
| Ähtäri II, FI0037R | 358 | 0.79 | 23.8 | 24.8 | -0.04 | 5.4 | 92 | 0.78 | 23.6 | 22.3 | 0.06 | 4.4 |
| Virolahti II, FI0017R | 358 | 0.70 | 25.0 | 26.6 | -0.06 | 7.0 | 91 | 0.77 | 26.1 | 24.5 | 0.07 | 5.5 |
| Utö, FI0009R | 362 | 0.74 | 28.1 | 31.1 | -0.09 | 6.4 | 90 | 0.68 | 28.9 | 34.6 | -0.17 | 7.6 |
| Rucava, LV0010R | 347 | 0.73 | 27.8 | 33.1 | -0.16 | 9.5 | 92 | 0.67 | 30.5 | 32.3 | -0.06 | 6.7 |
| Vilsandi, EE0011R | 361 | 0.80 | 28.8 | 31.7 | -0.09 | 6.1 | 90 | 0.83 | 31.1 | 35.7 | -0.13 | 6.3 |
| Lahemaa, EE0009R | 365 | 0.69 | 25.4 | 26.5 | -0.04 | 6.7 | 91 | 0.69 | 26.9 | 26.4 | 0.02 | 5.3 |
| Preila, LT0015R | 364 | 0.77 | 29.2 | 29.8 | -0.02 | 6.7 | 91 | 0.48 | 33.2 | 33.5 | -0.01 | 7.2 |





**Table 8.** Statistical evaluation of modelled $NO_2$ concentrations (in ppbv) with measurements of the EMEP monitoring network in the Baltic Sea region based on daily means for the entire year and for summer (JJA). Statistical indicators: mean values of model ($\mu_{Mod}$), mean values of observations ($\mu_{Obs}$), Spearman's correlation coefficient ($R_{Spr}$), normalized mean bias (NMB) and root mean square error of the modelled values (RMSE; in ppbv). All stations of the EMEP network located within the CD04 grid domain with available measurements were considered. $N$ is the number of the available daily mean measurements at the respective station.

| Station, Code | CD04 annual | | | | | | CD04 summer | | | | | |
| | $N$ | $R_{Spr}$ | $\mu_{Mod}$ | $\mu_{Obs}$ | NMB | RMSE | $N$ | $R_{Spr}$ | $\mu_{Mod}$ | $\mu_{Obs}$ | NMB | RMSE |
|---|---|---|---|---|---|---|---|---|---|---|---|---|
| Zingst, DE0009R | 355 | 0.73 | 3.8 | 3.4 | 0.10 | 2.04 | 88 | 0.58 | 2.9 | 2.4 | 0.22 | 1.50 |
| Keldsnor, DK0005R | 331 | 0.84 | 5.1 | 4.0 | 0.27 | 3.24 | 86 | 0.82 | 4.9 | 3.2 | 0.52 | 3.46 |
| Anholt, DK0008R | 283 | 0.81 | 3.7 | 2.6 | 0.42 | 2.67 | 92 | 0.80 | 3.6 | 2.0 | 0.83 | 2.91 |
| Risoe, DK0012R | 351 | 0.83 | 4.7 | 4.6 | 0.02 | 1.97 | 88 | 0.83 | 3.5 | 2.9 | 0.19 | 1.19 |
| Ähtäri II, FI0037R | 349 | 0.86 | 1.0 | 1.2 | -0.22 | 1.05 | 78 | 0.47 | 0.3 | 0.4 | -0.31 | 0.18 |
| Virolahti II, FI0017R | 365 | 0.65 | 2.1 | 2.6 | -0.18 | 2.60 | 92 | 0.54 | 1.5 | 1.5 | -0.01 | 0.90 |
| Hyytiälä, FI0050R | 142 | 0.90 | 1.3 | 1.8 | -0.28 | 1.63 | 32 | 0.70 | 0.3 | 0.4 | -0.10 | 0.19 |
| Utö, FI0009R | 331 | 0.71 | 1.9 | 1.6 | 0.18 | 1.26 | 87 | 0.66 | 2.1 | 1.4 | 0.48 | 1.50 |
| Rucava, LV0010R | 357 | 0.74 | 1.9 | 1.3 | 0.44 | 1.35 | 83 | 0.30 | 1.0 | 0.7 | 0.36 | 0.49 |
| Vilsandi, EE0011R | 340 | 0.69 | 1.6 | 1.3 | 0.24 | 1.12 | 89 | 0.49 | 1.4 | 0.8 | 0.71 | 1.44 |
| Lahemaa, EE0009R | 335 | 0.76 | 1.8 | 1.5 | 0.20 | 1.21 | 84 | 0.71 | 1.2 | 0.8 | 0.59 | 0.75 |
| Preila, LT0015R | 333 | 0.65 | 1.8 | 1.8 | 0.04 | 1.54 | 89 | 0.42 | 1.0 | 1.2 | -0.18 | 0.59 |




**Table 9.** Statistical evaluation of modelled $SO_2$ concentrations (in ppbv) with measurements of the EMEP monitoring network in the Baltic Sea region based on daily means for the entire year and for summer (JJA). Statistical indicators: mean values of model ($\mu_{Mod}$), mean values of observations ($\mu_{Obs}$), Spearman's correlation coefficient ($R_{Spr}$), normalized mean bias (NMB) and root mean square error of the modelled values (RMSE; in ppbv). All stations of the EMEP network located within the CD04 grid domain with available measurements were considered. $N$ is the number of the available daily mean measurements at the respective station.

| Station, Code | CD04 annual | | | | | | CD04 summer | | | | | |
|---|---|---|---|---|---|---|---|---|---|---|---|---|
| | $N$ | $R_{Spr}$ | $\mu_{Mod}$ | $\mu_{Obs}$ | NMB | RMSE | $N$ | $R_{Spr}$ | $\mu_{Mod}$ | $\mu_{Obs}$ | NMB | RMSE |
| Zingst, DE0009R | 366 | 0.54 | 1.21 | 0.87 | 0.40 | 0.98 | 92 | 0.54 | 0.75 | 0.80 | -0.07 | 0.37 |
| Anholt, DK0008R | 351 | 0.73 | 0.87 | 0.37 | 1.33 | 0.79 | 92 | 0.73 | 0.76 | 0.43 | 0.76 | 0.62 |
| Risoe, DK0012R | 350 | 0.70 | 1.03 | 0.42 | 1.46 | 0.91 | 92 | 0.73 | 0.68 | 0.37 | 0.81 | 0.45 |
| Ähtäri II, FI0037R | 50 | 0.61 | 0.50 | 0.35 | 1.15 | 0.40 | 12 | 0.04 | 0.19 | 0.07 | 1.84 | 0.17 |
| Virolahti II, FI0017R | 282 | 0.72 | 1.24 | 1.26 | -0.02 | 1.22 | 50 | 0.47 | 0.74 | 0.43 | 0.73 | 0.55 |
| Utö, FI0009R | 318 | 0.74 | 0.78 | 0.65 | 0.20 | 0.57 | 75 | 0.60 | 0.59 | 0.50 | 0.18 | 0.41 |
| Rucava, LV0010R | 359 | 0.58 | 1.25 | 0.48 | 1.62 | 1.50 | 92 | 0.29 | 0.60 | 0.35 | 0.71 | 0.52 |
| Vilsandi, EE0011R | 283 | 0.69 | 0.84 | 0.58 | 0.44 | 0.78 | 67 | 0.39 | 0.48 | 0.39 | 0.24 | 0.25 |
| Lahemaa, EE0009R | 270 | 0.57 | 1.13 | 1.17 | -0.04 | 1.45 | 75 | 0.42 | 0.61 | 0.58 | 0.05 | 0.41 |
| Preila, LT0015R | 302 | 0.57 | 1.29 | 0.60 | 1.15 | 1.47 | 92 | 0.30 | 0.55 | 0.24 | 1.28 | 0.43 |

**Table 10.** Statistical evaluation of modelled $PM_{2.5}$ concentrations (in $\mu g\,m^{-3}$) with measurements of the EMEP monitoring network in the Baltic Sea region based on daily means for the entire year and for summer (JJA). Statistical indicators: mean values of model ($\mu_{Mod}$), mean values of observations ($\mu_{Obs}$), Spearman's correlation coefficient ($R_{Spr}$), normalized mean bias (NMB) and root mean square error of the modelled values (RMSE; in ppbv). All stations of the EMEP network located within the CD04 grid domain with available measurements were considered. $N$ is the number of the available daily mean measurements at the respective station.

| Station, Code | CD04 annual | | | | | | CD04 summer | | | | | |
|---|---|---|---|---|---|---|---|---|---|---|---|---|
| | $N$ | $R_{Spr}$ | $\mu_{Mod}$ | $\mu_{Obs}$ | NMB | RMSE | $N$ | $R_{Spr}$ | $\mu_{Mod}$ | $\mu_{Obs}$ | NMB | RMSE |
| Råö, SE0014R | 351 | 0.49 | 4.9 | 5.6 | -0.13 | 4.4 | 87 | 0.36 | 2.5 | 4.9 | -0.50 | 3.5 |
| Diabla Gora, PL0005R | 365 | 0.82 | 7.8 | 13.8 | -0.44 | 8.5 | 92 | 0.70 | 2.5 | 8.7 | -0.71 | 6.8 |
| Vavihill, SE0011R | 252 | 0.55 | 6.3 | 8.2 | -0.23 | 5.2 | 84 | 0.62 | 2.8 | 6.5 | -0.58 | 4.5 |
| Aspvreten, SE0012R | 254 | 0.51 | 4.9 | 6.5 | -0.24 | 4.3 | 78 | 0.61 | 1.9 | 6.0 | -0.68 | 4.7 |
| Utö, FI0009R | 357 | 0.63 | 3.9 | 5.2 | -0.26 | 3.4 | 90 | 0.71 | 1.8 | 5.2 | -0.65 | 3.8 |
| Rucava, LV0010R | 327 | 0.29 | 6.8 | 10.8 | -0.37 | 10.4 | 78 | -0.19 | 2.5 | 7.5 | -0.66 | 6.5 |
| Vilsandi, EE0011R | 342 | 0.67 | 4.6 | 5.5 | -0.16 | 3.9 | 84 | 0.73 | 2.0 | 4.1 | -0.52 | 3.0 |
| Lahemaa, EE0009R | 343 | 0.60 | 6.0 | 5.5 | 0.09 | 4.6 | 87 | 0.32 | 1.8 | 3.6 | -0.50 | 2.7 |



**Table 11.** Future (2040) annual and seasonal nitrogen deposition amounts (kt N) to the seawater of the Baltic Sea and ship-related nitrogen deposition according to scenario "BAU 2040", taken from the CD04 grid. Values in brackets denote the change (in kt N) compared to 2012. Amounts refer to a Baltic Sea surface area of 431390 km², including the western part of Skagerrak.

| Nitrogen deposition | | Year | JFD | MAM | JJA | SON |
|---|---|---|---|---|---|---|
| All emissions | Oxidised | 35.7 (-58.8) | 10.9 (-12.2) | 5.6 (-10.5) | 6.9 (-16.2) | 12.3 (-19.8) |
| | Reduced | 52.9 (-11.6) | 8.1 (-1.0) | 15.3 (-3.1) | 13.9 (-3.6) | 15.6 (-3.9) |
| | Total | 88.6 (-70.3) | 19.0 (-13.2) | 20.9 (-13.6) | 20.8 (-19.8) | 27.9 (-23.7) |
| Ship emissions | Total | 4.9 (-17.6) | 0.8 (-3.1) | 0.9 (-3.4) | 1.8 (-6.7) | 1.4 (-4.4) |

**Table 12.** Summary of overall changes in future scenarios. Changes (in percent) on spatial average for all future scenarios compared to present-day (simulations with all emissions): annual means of $NO_2$, $SO_2$, $PM_{2.5}$ and the daily maximum $O_3$ within the 4-km resolution area (CD04 grid domains) and annual sum of nitrogen deposition to seawater.

| Scenario | $NO_2$ | $SO_2$ | $PM_{2.5}$ | $O_3$ daily max. | N depos. |
|---|---|---|---|---|---|
| "BAU 2040" | -72 | -61 | -37 | -4 | -44 |
| "NoNECA 2040" | -61 | -61 | -35 | -3 | -40 |
| "EEDI 2040" | -69 | -60 | -37 | -3 | -43 |

**Table 13.** Summary of ship contribution changes in future scenarios. Changes (in percent) on spatial average of the ship contributions for all future scenarios compared to present-day (simulations with all emissions): annual means of $NO_2$, $SO_2$, $PM_{2.5}$ and the daily maximum $O_3$ within the 4-km resolution area (CD04 grid domains) and annual sum of nitrogen deposition to seawater.

| Scenario | $NO_2$ | $SO_2$ | $PM_{2.5}$ | $O_3$ daily max. | N depos. |
|---|---|---|---|---|---|
| "BAU 2040" | -82 | -91 | -72 | -18 | -78 |
| "NoNECA 2040" | -55 | -90 | -48 | 31 | -46 |
| "EEDI 2040" | -75 | -88 | -61 | -1 | -69 |