# Peer review of "Impact of a nitrogen emission control area (NECA) on the future air quality and nitrogen deposition to seawater in the Baltic Sea region"

_Atmospheric Chemistry and Physics, 2018_

## Referee Comment (RC1) · Moussiopoulos (Referee) · 29 Nov 2018

This paper deals with the consequences of introducing a nitrogen emission control area in the Baltic Sea region. In addition, the authors analyze the influence of different fuel efficiency trends on their results. They assess both the impact on future air quality and on nitrogen deposition to seawater.

The manuscript is very well written and the results obtained convincingly presented. The authors apply a state-of-the-art methodology and use simulation models that are suitable for their study. The assumptions made are realistic and the input data appropriate. The illustrations included in the paper reflect successfully the key findings of the

analysis, and the conclusions are plausible.

As the only aspect worth commenting, the authors selected to work with an innermost grid of 4x4 km2. This resolution may be acceptable for regional scale simulations, yet it is clearly insufficient for predicting air quality in coastal urban areas, especially if such estimates are subsequently used to assess their impact to human health. Although it is understandable why the authors did not decide to increase the resolution to, say, 1 km, they should discuss the inevitable uncertainty associated with their intention to describe the air pollution situation in Baltic Sea harbours extending over hardly more than the assumed minimum cell surface area (16 km2).

---

## Referee Comment (RC2) · Anonymous Referee #2 · 30 Nov 2018

The paper is a modeling study to simulate the importance of present-day and future changes of shipping emissions on air quality and nitrogen deposition in and around the Baltic Sea and the North Sea. The analysis is solid and the conclusions are well supported. I recommend publication in ACP after my comments below are addressed.

My main comment is that the paper is quite dense with 13 tables and 14 figures, mostly multi-panel ones. While I acknowledge the efforts made by the authors to be thorough, some of the materials are better suited in the supplementary so that the main text is focused on the key messages. I suggest the following figures/tables and associated texts be placed in the supplementary; these are mostly model evaluations which can

be summarized in writing and then refer to specific figures/tables in the supplementary as backups to support the write-up. These are Figure 3 and 4 (precipitation evaluation), Table 3, Table 5 or Figure 5 (both show the model performance of nitrogen deposition; it is sufficient to keep one in the main text), and Table 7-10.

Introduction: the present write-up does not mention the importance of NOx as a precursor of tropospheric ozone. This should be added, as the paper also presents the effect of shipping emissions on ozone.

Pg6, line 8-14: The writing on the treatment of sub-grid clouds is confusing. First, does the CMAQ model used for the manuscript treat sub-grid cloud or not? The last sentence seems to indicate it does not. If so, then the preceding sentences on the sub-grid clouds should be removed and replaced by a simple statement saying that sub-grid cloud treatment available in the standard CMAQ model is not used. Second, I am not convinced that the 4km x 4km resolution is sufficiently fine to resolve convective clouds. Do you have references or model simulations to support this?

Pg1, L10: "emission" should be "emissions"

Pg2, l15: Spell out the full name for MARPOL when it first appears in the paper

Pg2, l16: ";" should be ","

---

## Author Comment (AC2) · 11 Jan 2019

We thank Referee #2 for positive evaluation of the manuscript. Following the Reviewer's remarks, the evaluation parts on precipitation and nitrogen deposition have been substantially condensed in the revised manuscript. We have addressed the issues brought up below, with specific pointers to changed parts of the manuscript.

**1. My main comment is that the paper is quite dense with 13 tables and 14 figures, mostly multi-panel ones. While I acknowledge the efforts made by the authors to be thorough, some of the materials are better suited in the supplementary so that the main text is focused on the key messages. I suggest the following**

[Figure]

**figures/tables and associated texts be placed in the supplementary; these are mostly model evaluations which can be summarized in writing and then refer to specific figures/tables in the supplementary as backups to support the write-up. These are Figure 3 and 4 (precipitation evaluation), Table 3, Table 5 or Figure 5 (both show the model performance of nitrogen deposition; it is sufficient to keep one in the main text), and Table 7-10.**

Reply: Figure 3, Figure 4 and Table 3 have been moved to the Supplement. Table 4 (WNO3) and Table 5 (WNH4) together show the evaluation of the wet deposition of nitrogen. Both have been moved to the Supplement. Figure 5 has been kept in the main text because it illustrates the differences between CD16 and CD04 resolutions. Table 7, 8, 9, and 10 have been moved to the Supplement. Figures and Tables have been renumbered accordingly. Associated text has been arranged into three new sections in the Supplement that describe the methodology and results of the evaluation of modelled data with observations: Sect. S1 "Methodology and results for the evaluation of modelled precipitation", Sect. S2 "Methodology and results for the evaluation of modelled wet deposition of nitrogen", and Sect. S3 "Methodology and results for the evaluation of modelled air pollutant concentrations".

**2. Introduction: the present write-up does not mention the importance of NOx as a precursor of tropospheric ozone. This should be added, as the paper also presents the effect of shipping emissions on ozone.**

Reply: The following has been added on page 1, Line 26: "The atmospheric transformation of emitted NOx from shipping is especially relevant for the formation of ozone (Eyring et al., 2010). Shipping emissions are estimated to play an important role on ozone ($O_3$) levels compared to the road transport sector near the coastal zone in Europe (Tagaris et al., 2017). A regional impact study by Huszar et al. (2010) found that the contribution of shipping emissions to surface NOx levels causes an increase of surface $O_3$ by up to 4–6 ppbv over the eastern Atlantic and western Europe. $O_3$ can damage vegetation, reduce plant primary productivity and agricultural crop yields

(Chuwah et al., 2015) and is also a serious concern for human health (EEA, 2015)."

**3. a) Pg6, line 8-14: The writing on the treatment of sub-grid clouds is confusing. First, does the CMAQ model used for the manuscript treat sub-grid cloud or not? The last sentence seems to indicate it does not. If so, then the preceding sentences on the sub-grid clouds should be removed and replaced by a simple statement saying that sub-grid cloud treatment available in the standard CMAQ model is not used.**

Reply: CMAQ simulates the presence of both resolved and sub-grid clouds and their effects on atmospheric chemistry. The Meteorology-Chemistry Interface Processor of CMAQ, MCIP, diagnoses for each horizontal grid cell the cloud coverage, cloud base and top, and the average liquid water content in the cloud using a series of simple algorithms based on a relative humidity threshold (Otte and Pleim, 2010). These cloud algorithms are described in detail in Byun et al. (1999). The transport processes in the atmosphere primarily consist of advection and diffusion, except for the mixing of pollutants by the parameterized sub-grid clouds (Byun et al., 1999). CMAQ first determines the presence of resolved clouds (clouds resolved in the output of the meteorological model) based on the total liquid water mixing ratio in each vertical layer. Next CMAQ diagnosis the presence of sub-grid precipitating convective clouds and determines the fractional cloud cover of sub-grid non-precipitating clouds. If the convective clouds are resolved in the output of the meteorological model, the diagnosis of sub-grid clouds in CMAQ becomes unnecessary and should be turned off. In CMAQ v5.0.1., the version used in the present study, the cloud module was updated to only simulate sub-grid clouds when the meteorological driver has used a convective cloud parameterization (https://www.airqualitymodeling.org/index.php/CMAQ_version_5.0_(February_2012_release)_Technical_Documentation). Switching off the sub-grid cloud diagnosis in CMAQ circumvents the known limitations of CMAQ in correctly determining the cloud cover of sub-grid non-precipitating clouds. In the presented simulations with CMAQ v5.0.1, the sub-grid clouds were simulated for the 64-km grid resolution and the 16-km resolution, but not for the 4-km resolution because the convective clouds have been resolved in the output of COSMO-CLM for the high-resolution grid.

To avoid ambiguities, text on Page 6, Lines 8-14 has been revised as follows: "Three types of clouds are modelled in CMAQ: sub-grid convective precipitation clouds, sub-grid non-precipitating clouds and grid-resolved clouds. CMAQ simulates the aqueous phase chemistry in all cloud types. For the two types of sub-grid clouds, the cloud module in CMAQ vertically redistributes pollutants and calculates in-cloud and precipitation scavenging. Since the meteorological model provides information about the grid-resolved clouds, CMAQ subsequently does not apply further cloud dynamics for this cloud type. Sub-grid clouds are only simulated in CMAQ when the meteorological driver uses a convective cloud parameterization. Hence sub-grid clouds are treated by CMAQ on the coarser outer resolution grids (64-km and 16-km) but not on the $4 \times 4$ km$^2$ model domain because the convective clouds are resolved for the fine grid resolution by the meteorological model."

**3. b) Second, I am not convinced that the 4km x 4km resolution is sufficiently fine to resolve convective clouds. Do you have references or model simulations to support this?**

Reply: Meteorological fields that were used in the CMAQ simulation on the $4 \times 4$ km$^2$ resolution grid are from COSMO-CLM output with a horizontal resolution of 0.025 degree (see Page 3, Line 26 and Page 6, Line 18) which due to the rotated lat-lon coordinate system corresponds to a resolution of 2.8 km. At this resolution convective-scale circulations are resolved. With decreasing horizontal grid spacing, the convective parameterizations in meteorological models become more and more inappropriate and scientifically questionable given the underlying assumptions. Previous studies demonstrated that a 4-km resolution of the Weather Research and Forecasting (WRF) model with Advanced Research WRF (ARW) dynamics core (Skamarock et al., 2005, 2008), which explicitly resolves convection, gives better results for precipitation forecast compared to the 12-km resolution operational North American Mesoscale model (NAM)

model, which employs convective parameterization (Kain et al., 2006; Weisman et al., 2008; Schwartz et al., 2009; Coniglio et al., 2013). Overall, 4-km convection-allowing configurations enable a reasonable evolution of the convective overturning process even though a 4-km grid is too coarse to fully capture convective-scale circulations (Schwartz et al., 2009).

**4. Pg1, L10: "emission" should be "emissions"**

Reply: Changed.

**5. Pg2, l15: Spell out the full name for MARPOL when it first appears in the paper**

Reply: Changed.

**6. Pg2, l16: ";" should be ","**

Reply: Changed.

References:

Byun, D. and Ching, J.: Science Algorithms of the EPA Models-3 Community Multiscale Air Quality Modeling System, EPA/600/r-99/030, US Environmental Protection Agency, Office of Research and Development, Washington DC, 1999.

Chuwah, C., van Noije, T., van Vuuren, D. P., Stehfest, E., and Hazeleger, W., 2015. Global impacts of surface ozone changes on crop yields and land use, Atmos. Environ., 106, 11–23. http://dx.doi.org/10.1016/j.atmosenv.2015.01.062, 2015.

Coniglio, M. C., Correia Jr., J., Marsh, P. T., and Kong, F.: Verification of Convection-Allowing WRF Model Forecasts of the Planetary Boundary Layer Using Sounding Observations, Weather and Forecasting, 28(3), doi:10.1175/WAF-D-12-00103.1, 2013.

EEA: Air quality in Europe - 2015 Report, European Environment Agency, EEA Report. No. 5/2015, Copenhagen, 57 pp., 2015.

Eyring, V., Isaksen, I., Berntsen, T., Collins, W., Corbett, J., Endresen, Ø., Grainger,

R., Moldanová, J., Schlager, H., and Stevenson, D.: Transport impacts on atmosphere and climate: shipping, Atmos. Environ., 44, 4735−4771, 2010.

Huszar, P., Cariolle, D., Paoli, R., Halenka, T., Belda, M., Schlager, H., Miksovsky, J., and Pisoft, P.: Modeling the regional impact of ship emissions on NOx and ozone levels over the Eastern Atlantic and Western Europe using ship plume parameterization. Atmos. Chem. Phys., 10, 6645–6660, doi:10.5194/acp-10-6645-2010, 2010.

Kain, J. S., Weiss, S. J., Levit, J. J., Baldwin, M. E., and Bright, D. R.: Examination of convection-allowing configurations of the WRF model for the prediction of severe convective weather: the SPC/NSSL Spring Program 2004, Weather and Forecasting, 21(2), 167−181, 2006.

Otte, T. L. and Pleim, J. E.: The Meteorology-Chemistry Interface Processor (MCIP) for the CMAQ modeling system: updates through MCIPv3.4.1, Geosci. Model Dev., 3, 243−256, doi:10.5194/gmd-3-243-2010, 2010.

Schwartz, C. S., Kain, J. S., Weiss, S. J., Xue, M., Bright, D. R., Kong, F., Thomas, K. W., Levit, J. J., and Coniglio, M. C.: Next-day convection-allowing WRF model guidance: A second look at 2 vs. 4 km grid spacing, Mon. Wea. Rev., 137, 3351–3372, 2009.

Skamarock, W. C., Klemp, J. B., Dudhia, J., Gill, D. O., Barker, D. M., Wang, W., and Powers, J. G.: A description of the Advanced Research WRF Version 2, NCAR Tech Note, NCAR/TN-468+STR, 88 pp., 2005.

Skamarock, W. C., Klemp, J. B., Dudhia, J., Gill, D. O., Barker, D. M., Duda, M. G., Huang, X.-Y., Wang, W., and Powers, J. G.: A Description of the Advanced Research WRF Version 3, NCAR Technical note, NCAR/TN-475+STR, available at: http://www.mmm.ucar.edu/wrf/users/docs/arw_v3.pdf, 2008.

Tagaris, E., Stergiou, I., and Sotiropoulou, R.-E. P.: Impact of shipping emissions on ozone levels over Europe: assessing the relative importance of the Standard Nomen-

clature for Air Pollution (SNAP) categories, Environ. Sci. Pollut. Res., 24, 14903–14909, doi:10.1007/s11356-017-9046-x, 2017.

Weisman, M. L., Davis, C., Wang, W., Manning, K. W., and Klemp, J. B.: Experiences with 0-36-h explicit convective forecasts with the WRF-ARW model, Weather and Forecasting, vol. 23(3), 407−437, 2008.

---

## Author Response (AR1)

**Changes to manuscript ms-nr acp-2018-1107**

**Impact of a nitrogen emission control area (NECA) on the future air quality and nitrogen deposition to seawater in the Baltic Sea region**

Matthias Karl (1,*), Johannes Bieser (1), Beate Geyer (1), Volker Matthias (1), Jukka-Pekka Jalkanen (2), Lasse Johansson (2), Erik Fridell (3)

[1] Institute of Coastal Research, Helmholtz-Zentrum Geesthacht, 21502 Geesthacht, Germany.

[2] Atmospheric Composition Research, Finnish Meteorological Institute, P.O. Box 503, FI-00101 Helsinki, Finland.

[3] IVL Swedish Environmental Research Institute, P.O. Box, SE 40014, Gothenburg, Sweden.

**Dear Dr Huan Liu,**

We highly appreciate the reviews of our manuscript ms-nr acp-2018-1107 that we received from Prof Nicolas Moussiopoulos and one anonymous referee. We have replied to their comments in the Open Discussion. We have addressed all specific comments in the revised manuscript as will be described below. We carefully considered the concerns of the anonymous referee and the comment of Prof Nicolas Moussiopoulos in our revision of the manuscript.

Below follows: (1) the point-by-point replies to the two reviewers, (2) a list of relevant changes in the manuscript, and (3) the revised manuscript with changes highlighted.

Our responses to reviewers have been written in blue font.

Figure, table, section, and page numbers in the replies below refer to the original manuscript. The revised manuscript with changes highlighted has been sent along with this response.

**Prof Moussiopoulos**

1. The manuscript is very well written and the results obtained convincingly presented. The authors apply a state-of-the-art methodology and use simulation models that are suitable for their study. The assumptions made are realistic and the input data appropriate. The illustrations included in the paper reflect successfully the key findings of the analysis, and the conclusions are plausible.

We thank the Reviewer for their assessment of the scope and methodology of the manuscript.

2. As the only aspect worth commenting, the authors selected to work with an innermost grid of 4x4 km2. This resolution may be acceptable for regional scale simulations, yet it is clearly insufficient for predicting air quality in coastal urban areas, especially if such estimates are subsequently used to assess their impact to human health. Although it is understandable why the authors did not decide to increase the resolution to, say, 1 km, they should discuss the inevitable uncertainty associated with their intention to describe the air pollution situation in Baltic Sea harbours extending over hardly more than the assumed minimum cell surface area (16 km2).

The presented study makes no claims about predicting air quality and human health impacts within coastal urban areas. However, results from our study can be used for the regional scale assessment of health impacts due to shipping emissions and the relative change of these impacts in the future (2040) when the NECA is introduced in 2021, given that regulations are implemented as outlined in the future ship emission scenarios. Brandt et al. (2013b) used an integrated model system, Economic Valuation of Air pollution (EVA) that integrates a regional scale atmospheric chemistry transport model, to examine the relative health-related external costs in Denmark from international ship traffic in the Baltic Sea and the North Sea. The finest horizontal resolution applied in that study was 16.67 km covering the North Sea region and parts of the Baltic Sea region (Brandt et al., 2013a). Compared to Brandt et al. (2013b) the present study allows for a better resolution of the coastal areas and the ship traffic. Ideally, a resolution of 1 km should be used for resolving the urban increment (Schaap et al., 2015) in the coastal areas. However, operating CMAQ at a grid resolution of 1 km is not feasible for the extent of the Baltic Sea region because of the data demands and the enormous increase of computational time. We refer to manuscripts in preparation for the Special Issue "Shipping and the Environment - From Regional to Global Perspectives" that deal with the impact of shipping on the urban air quality in harbour cities of the Baltic Sea. The model output from the regional scale simulations with CMAQ is used as boundary conditions for the urban domains to simulate the present-day air quality in the harbour cities Rostock, Gdansk and Riga (Ramacher et al., 2019b) and the air quality in Gothenburg in 2040 (Ramacher et al., 2019a).

The following has been added in the Conclusions (page 29, Line 21) with respect to the limitations for use of the regional scale model data for health impact assessments in urban areas:

"Use of the presented model data for health impact assessment in the densely populated coastal areas of the Baltic Sea region is connected to uncertainties arising from limitations of the chosen grid resolution. Despite the fine spatial resolution of the inner-most model grid, the concentration gradients between urban areas and their surroundings (urban increment) and within harbour cities are not adequately resolved by the simulations due to the large spatial and temporal variability of emissions in urban areas. Ideally, a grid length of 1 km should be chosen to resolve the urban increments (Schaap et al., 2015) in the coastal areas. However, a finer resolution brings along the need for more accurate emission data in the urban areas, which is challenging because the compilation of urban emission inventories requires specific information for each emitting sector (Guevara et al, 2016)."

Brandt, J., Silver, J. D., Christensen, J. H., Andersen, M. S., Bønløkke, J. H., Sigsgaard, T., Geels, C., Gross, A., Hansen, A. B., Hansen, K. M., Hedegaard, G. B., Kaas, E., and Frohn, L. M.: Contribution from the ten major emission sectors in Europe and Denmark to the health-cost externalities of air pollution using the EVA model system – an integrated modelling approach, Atmos. Chem. Phys., 13, 7725–7746, doi:10.5194/acp-13-7725-2013, 2013a.

Brandt, J., Silver, J. D., Christensen, J. H., Andersen, M. S., Bønløkke, J. H., Sigsgaard, T., Geels, C., Gross, A., Hansen, A. B., Hansen, K. M., Hedegaard, G. B., Kaas, E., and Frohn, J. M.: Assessment of past, present and future health-cost externalities of air pollution in Europe and the contribution from international ship traffic using the EVA model system, Atmos. Chem. Phys., 13, 7747–7764, doi:10.5194/acp-13-7747-2013, 2013b.

Guevara, M., Lopez-Aparicio, S., Cuvelier, C., Tarrason, L., Clappier, A., and Thunis, P.: A benchmarking tool to screen and compare bottom 25 up and top-down atmospheric emission inventories, Air Qual. Atmos. Health, 10, 627–642, https://doi.org/10.1007/s11869-016-0456-6, 2016.

Ramacher, M. O. P., Tang, L., Moldanová, J., Matthias, V., Karl, M., and Johansson, L.: The impact of ship emissions on air quality and human health in the Gothenburg area - Part II: Scenarios for 2040, manuscript in prep., Atmos. Chem. Phys., 2019a.

Ramacher, M. O. P., Karl, M., Jalkanen, J.-P., and Johansson, L.: Population exposure to emissions from ships in Baltic Sea harbour cities, manuscript in prep., Atmos. Chem. Phys., 2019b.

Schaap, M., Cuvelier, C., Hendriks, C., Bessagnet, B., Baldasano, J. M., Colette, A., Thunis, P., Karam, D., Fagerli, H., Graff, A., Kranenburg, R., Nyíri, A., Pay, M. T., Rouïl, L., Schulz, M., Simpson, D., Stern, R., Terrenoire, E., and Wind, P.: Performance of European chemistry transport models as function of horizontal resolution, Atmos. Environ., 112, 90–105, http://dx.doi.org/10.1016/j.atmosenv.2015.04.003, 2015.

**Referee #2**

1. My main comment is that the paper is quite dense with 13 tables and 14 figures, mostly multi-panel ones. While I acknowledge the efforts made by the authors to be thorough, some of the materials are better suited in the supplementary so that the main text is focused on the key messages. I suggest the following figures/tables and associated texts be placed in the supplementary; these are mostly model evaluations which can be summarized in writing and then refer to specific figures/tables in the supplementary as backups to support the write-up. These are Figure 3 and 4 (precipitation evaluation), Table 3, Table 5 or Figure 5 (both show the model performance of nitrogen deposition; it is sufficient to keep one in the main text), and Table 7-10.

Figure 3, Figure 4 and Table 3 have been moved to the Supplement. Table 4 (WNO3) and Table 5 (WNH4) together show the evaluation of the wet deposition of nitrogen. Both have been moved to the Supplement. Figure 5 has been kept in the main text because it illustrates the differences between CD16 and CD04 resolutions. Table 7, 8, 9, and 10 have been moved to the Supplement. Figures and Tables have been renumbered accordingly. Associated text has been arranged into three new sections in the Supplement that describe the methodology and results of the evaluation of modelled data with observations: Sect. S1" Methodology and results for the evaluation of modelled precipitation", Sect. S2 "Methodology and results for the evaluation of modelled wet deposition of nitrogen", and Sect. 3 "Methodology and results for the evaluation of modelled air pollutant concentrations".

2. Introduction: the present write-up does not mention the importance of NOx as a precursor of tropospheric ozone. This should be added, as the paper also presents the effect of shipping emissions on ozone.

The following has been added on page 2, Line 26:

"The atmospheric transformation of emitted NOx from shipping is especially relevant for the formation of ozone (Eyring et al., 2010). Shipping emissions are estimated to play an important role on ozone (O3) levels compared to the road transport sector near the coastal zone in Europe (Tagaris et al., 2017). A regional impact study by Huszar et al. (2010) found that the contribution of shipping emissions to surface NOx levels causes an increase of surface O3 by up to 4–6 ppbv over the eastern Atlantic and western Europe. O3 can damage vegetation, reduce plant primary productivity and agricultural crop yields (Chuwah et al., 2015) and is also a serious concern for human health (EEA, 2015)."

Chuwah, C., van Noije, T., van Vuuren, D. P., Stehfest, E., and Hazeleger, W., 2015. Global impacts of surface ozone changes on crop yields and land use, Atmos. Environ., 106, 11–23. http://dx.doi.org/10.1016/j.atmosenv.2015.01.062, 2015.

EEA: Air quality in Europe - 2015 Report, European Environment Agency, EEA Report. No. 5/2015, Copenhagen, 57 pp., 2015.

Eyring, V., Isaksen, I., Berntsen, T., Collins, W., Corbett, J., Endresen, Ø., Grainger, R., Moldanová, J., Schlager, H., and Stevenson, D.: Transport impacts on atmosphere and climate: shipping, Atmos. Environ., 44, 4735–4771, 2010.

Huszar, P., Cariolle, D., Paoli, R., Halenka, T., Belda, M., Schlager, H., Miksovsky, J., and Pisoft, P.: Modeling the regional impact of ship emissions on NOx and ozone levels over the Eastern Atlantic and Western Europe using ship plume parameterization. Atmos. Chem. Phys., 10, 6645–6660, doi:10.5194/acp-10-6645-2010, 2010.

3.  a) Pg6, line 8-14: The writing on the treatment of sub-grid clouds is confusing. First, does the CMAQ model used for the manuscript treat sub-grid cloud or not? The last sentence seems to indicate it does not. If so, then the preceding sentences on the sub-grid clouds should be removed and replaced by a simple statement saying that sub-grid cloud treatment available in the standard CMAQ model is not used.

CMAQ simulates the presence of both resolved and sub-grid clouds and their effects on atmospheric chemistry. The Meteorology-Chemistry Interface Processor of CMAQ, MCIP, diagnoses for each horizontal grid cell the cloud coverage, cloud base and top, and the average liquid water content in the cloud using a series of simple algorithms based on a relative humidity threshold (Otte and Pleim, 2010). These cloud algorithms are described in detail in Byun et al. (1999). The transport processes in the atmosphere primarily consist of advection and diffusion, except for the mixing of pollutants by the parameterized sub-grid clouds (Byun et al., 1999). CMAQ first determines the presence of resolved clouds (clouds resolved in the output of the meteorological model) based on the total liquid water mixing ratio in each vertical layer. Next CMAQ diagnosis the presence of sub-grid precipitating convective clouds and determines the fractional cloud cover of sub-grid non-precipitating clouds. If the convective clouds are resolved in the output of the meteorological model, the diagnosis of sub-grid clouds in CMAQ becomes unnecessary and should be turned off. In CMAQ v5.0.1., the version used in the present study, the cloud module was updated to only simulate sub-grid clouds when the meteorological driver has used a convective cloud parameterization (https://www.airqualitymodeling.org/index.php/CMAQ_version_5.0_(February_2012_release)_Technical_Documentation). Switching off the sub-grid cloud diagnosis in CMAQ circumvents the known limitations of CMAQ in correctly determining the cloud cover of sub-grid non-precipitating clouds. In the presented simulations with CMAQ v5.0.1, the sub-grid clouds were simulated for the 64-km grid resolution and the 16-km resolution, but not for the 4-km resolution because the convective clouds have been resolved in the output of COSMO-CLM for the high-resolution grid. To avoid ambiguities, text on Page 6, Lines 8-14 has been revised as follows:

"Three types of clouds are modelled in CMAQ: sub-grid convective precipitation clouds, sub-grid non-precipitating clouds and grid-resolved clouds. CMAQ simulates the aqueous phase chemistry in all cloud types. For the two types of sub-grid clouds, the cloud module in CMAQ vertically redistributes pollutants and calculates in-cloud and precipitation scavenging. Since the meteorological model provides information about the grid-resolved clouds, CMAQ subsequently does not apply further cloud dynamics for this cloud type. Sub-grid clouds are

only simulated in CMAQ when the meteorological driver uses a convective cloud parameterization. Hence sub-grid clouds are treated by CMAQ on the coarser outer resolution grids (64-km and 16-km) but not on the 4 × 4 km2 model domain because the convective clouds are resolved for the fine grid resolution by the meteorological model."

Byun, D. and Ching, J.: Science Algorithms of the EPA Models-3 Community Multiscale Air Quality Modeling System, EPA/600/r-99/030, US Environmental Protection Agency, Office of Research and Development, Washington DC, 1999.

Otte, T. L. and Pleim, J. E.: The Meteorology-Chemistry Interface Processor (MCIP) for the CMAQ modeling system: updates through MCIPv3.4.1, Geosci. Model Dev., 3, 243–256, doi:10.5194/gmd-3-243-2010, 2010.

3. b) Second, I am not convinced that the 4km x 4km resolution is sufficiently fine to resolve convective clouds. Do you have references or model simulations to support this?

Meteorological fields that were used in the CMAQ simulation on the 4x4 km2 resolution grid are from COSMO-CLM output with a horizontal resolution of 0.025 degree (see Page 3, Line 26 and Page 6, Line 18) which due to the rotated lat-lon coordinate system corresponds to a resolution of 2.8 km. At this resolution convective-scale circulations are resolved. With decreasing horizontal grid spacing, the convective parameterizations in meteorological models become more and more inappropriate and scientifically questionable given the underlying assumptions. Previous studies demonstrated that a 4-km resolution of the Weather Research and Forecasting (WRF) model with Advanced Research WRF (ARW) dynamics core (Skamarock et al., 2005, 2008), which explicitly resolves convection, gives better results for precipitation forecast compared to the 12-km resolution operational North American Mesoscale model (NAM) model, which employs convective parameterization (Kain et al., 2006; Weisman et al., 2008; Schwartz et al., 2009; Coniglio et al., 2013). Overall, 4-km convection-allowing configurations enable a reasonable evolution of the convective overturning process even though a 4-km grid is too coarse to fully capture convective-scale circulations (Schwartz et al., 2009).

Coniglio, M. C., Correia Jr., J., Marsh, P. T., and Kong, F.: Verification of Convection-Allowing WRF Model Forecasts of the Planetary Boundary Layer Using Sounding Observations, Weather and Forecasting, 28(3), doi:10.1175/WAF-D-12-00103.1, 2013.

Kain, J. S., Weiss, S. J., Levit, J. J., Baldwin, M. E., and Bright, D. R.: Examination of convection-allowing configurations of the WRF model for the prediction of severe convective weather: the SPC/NSSL Spring Program 2004, Weather and Forecasting, 21(2), 167–181, 2006.

Schwartz, C. S., Kain, J. S., Weiss, S. J., Xue, M., Bright, D. R., Kong, F., Thomas, K. W., Levit, J. J., and Coniglio, M. C.: Next-day convection-allowing WRF model guidance: A second look at 2 vs. 4 km grid spacing, Mon. Wea. Rev., 137, 3351–3372, 2009.

Skamarock, W. C., Klemp, J. B., Dudhia, J., Gill, D. O., Barker, D. M., Wang, W., and Powers, J. G.: A description of the Advanced Research WRF Version 2, NCAR Tech Note, NCAR/TN-468+STR, 88 pp., 2005.

Skamarock, W. C., Klemp, J. B., Dudhia, J., Gill, D. O., Barker, D. M., Duda, M. G., Huang, X.-Y., Wang, W., and Powers, J. G.: A Description of the Advanced Research WRF Version 3, NCAR Technical note, NCAR/TN-475+STR, available at: http://www.mmm.ucar.edu/wrf/users/docs/arw_v3.pdf, 2008.

Weisman, M. L., Davis, C., Wang, W., Manning, K. W., and Klemp, J. B.: Experiences with 0-36-h explicit convective forecasts with the WRF-ARW model, Weather and Forecasting, vol. 23(3), 407–437, 2008.

4. Pg1, L10: "emission" should be "emissions"

Changed.

5. Pg2, l15: Spell out the full name for MARPOL when it first appears in the paper

Changed.

6. Pg2, l16: ";" should be ","

Changed.

**List of relevant changes in the ms**

**Relevant text changes:**

**1. Introduction**

We address the relevance of $NO_X$ as a precursor of tropospheric ozone as suggested by Referee #2 by adding the following text on Page 2, Line 26:

"The atmospheric transformation of emitted $NO_X$ from shipping is especially relevant for the formation of ozone (Eyring et al., 2010). Shipping emissions are estimated to play an important role on ozone ($O_3$) levels compared to the road transport sector near the coastal zone in Europe (Tagaris et al., 2017). A regional impact study by Huszar et al. (2010) found that the contribution of shipping emissions to surface $NO_X$ levels causes an increase of surface $O_3$ by up to 4–6 ppbv over the eastern Atlantic and western Europe. $O_3$ can damage vegetation, reduce plant primary productivity and agricultural crop yields (Chuwah et al., 2015) and is also a serious concern for human health (EEA, 2015)."

**2. Section 2.1 "CMAQ Model Description"**

As noted by Referee #2, the description of the sub-grid cloud treatment in the CMAQ simulations is confusing. In the presented simulations with CMAQ v5.0.1, the sub-grid clouds were simulated for the 64-km grid resolution and the 16-km resolution, but not for the 4-km resolution because the convective clouds have been resolved in the output of COSMO-CLM for the high-resolution grid. The description of the treatment of sub-grid clouds and grid-resolved clouds in the CMAQ simulations has been revised (Page 6, Lines 8-14) as follows:

"Three types of clouds are modelled in CMAQ: sub-grid convective precipitation clouds, sub-grid non-precipitating clouds and grid-resolved clouds. CMAQ simulates the aqueous phase chemistry in all cloud types. For the two types of sub-grid clouds, the cloud module in CMAQ vertically redistributes pollutants and calculates in-cloud and precipitation scavenging. Since the meteorological model provides information about the grid-resolved clouds, CMAQ subsequently does not apply further cloud dynamics for this cloud type. Sub-grid clouds are only simulated in CMAQ when the meteorological driver uses a convective cloud parameterization. Hence sub-grid clouds are treated by CMAQ on the coarser outer resolution grids (64-km and 16-km) but not on the $4 \times 4$ $km^2$ model domain because the convective clouds are resolved for the fine grid resolution by the meteorological model."

**3. Section 4.1 "Present-day nitrogen deposition"**

Following the main comment of Referee #2, large text parts, several tables and figures of Section 4.1 "Present-day nitrogen deposition" have been moved to the Supplementary Information in order to strengthen the focus on the key messages of the manuscript. Section

4.1.1 "Comparison of the modelled precipitation with observations" has been condensed and related figures and tables (Figure 3, Figure 4 and Table 3) have been moved to the Supplement. The description of the methodology for evaluation of modelled precipitation and the detailed results are now given in Section S1 "Methodology and results for the evaluation of modelled precipitation" of the Supplement. Section 4.1.2 has been renamed to "Comparison of the modelled wet deposition of nitrogen with observations". Table 4 (WNO3) and Table 5 (WNH4) have been moved to the Supplement. Figure 5 has been kept in the main text because it illustrates the differences between CD16 and CD04 resolutions. The description of the methodology for evaluation of nitrogen wet deposition are now given in Section S2 "Methodology and results for the evaluation of modelled wet deposition of nitrogen" of the Supplement.

**4.     Section 4.2 "Present-day air quality"**

Following the main comment of Referee #2, several text parts and the tables of Section 4.2 "Present-day air quality" have been moved to the Supplementary Information in order to strengthen the focus on the key messages of the manuscript. Table 7, 8, 9, and 10 have been moved to the Supplement. The description of the methodology for evaluation of modelled concentrations of air pollutants and the detailed results are now given in Section S3 "Methodology and results for the evaluation of modelled air pollutant concentrations" of the Supplement.

Figures and tables in the revised manuscript have been renumbered accordingly, see below.

**5.     Conclusions**

To address the concern of Prof Moussiopoulos about the use of model data from the $4 \times 4$ km$^2$ resolution grid for health impact assessment in urban areas of the Baltic Sea region, the following text has been added in the Conclusions (Page 29, Line 21):

"Use of the presented model data for health impact assessment in the densely populated coastal areas of the Baltic Sea region is connected to uncertainties arising from limitations of the chosen grid resolution. Despite the fine spatial resolution of the inner-most model grid, the concentration gradients between urban areas and their surroundings (urban increment) and within harbour cities are not adequately resolved by the simulations due to the large spatial and temporal variability of emissions in urban areas. Ideally, a grid length of 1 km should be chosen to resolve the urban increments (Schaap et al., 2015) in the coastal areas. However, a finer resolution brings along the need for more accurate emission data in the urban areas, which is challenging because the compilation of urban emission inventories requires specific information for each emitting sector (Guevara et al, 2016)."

**Tables:**

Table 3.

The table has been moved to the Supplementary Information and is now Table S2.

Table 4.

The table has been moved to the Supplementary Information and is now Table S3.

Table 5.

The table has been moved to the Supplementary Information and is now Table S4.

Table 6.

The table has been renumbered as Table 3.

Table 7.

The table has been moved to the Supplementary Information and is now Table S7.

Table 8.

The table has been moved to the Supplementary Information and is now Table S8.

Table 9.

The table has been moved to the Supplementary Information and is now Table S9.

Table 10.

The table has been moved to the Supplementary Information and is now Table S10.

Table 11.

The table has been renumbered as Table 4.

Table 12.

The table has been renumbered as Table 5.

Table 14.

The table has been renumbered as Table 6.

**Figures:**

Figure 3.

The figure has been moved to the Supplementary Information and is now Figure S1.

Figure 4.

The figure has been moved to the Supplementary Information and is now Figure S2.

Figure 5.

The figure has been renumbered as Figure 3.

Figure 6.

The figure has been renumbered as Figure 4.

Figure 7.

The figure has been renumbered as Figure 5.

Figure 8.

The figure has been renumbered as Figure 6.

Figure 9.

The figure has been renumbered as Figure 7.

Figure 10.

The figure has been renumbered as Figure 8.

Figure 11.

The figure has been renumbered as Figure 9.

Figure 12.

The figure has been renumbered as Figure 10.

Figure 13.

The figure has been renumbered as Figure 11.

Figure 14.

The figure has been renumbered as Figure 12.

[revised manuscript text omitted]